# On the insufficiency of existing momentum schemes for Stochastic Optimization

**Rahul Kidambi**[‡], **Praneeth Netrapalli**[2], **Prateek Jain**[2] **and Sham M. Kakade**[1]

[1] University of Washington Seattle    [2] Microsoft Research India

`rkidambi@uw.edu`, {`praneeth`, `prajain`}`@microsoft.com`,
`sham@cs.washington.edu`

## Abstract

Momentum based stochastic gradient methods such as heavy ball (HB) and Nesterov's accelerated gradient descent (NAG) method are widely used in practice for training deep networks and other supervised learning models, as they often provide significant improvements over stochastic gradient descent (SGD). Rigorously speaking, "fast gradient" methods have provable improvements over gradient descent only for the deterministic case, where the gradients are exact. In the stochastic case, the popular explanations for their wide applicability is that when these fast gradient methods are applied in the stochastic case, they partially mimic their *exact* gradient counterparts, resulting in some practical gain. This work provides a counterpoint to this belief by *proving* that there exist simple problem instances where these methods *cannot* outperform SGD despite the best setting of its parameters. These negative problem instances are, in an informal sense, generic; they do not look like carefully constructed pathological instances. These results suggest (along with empirical evidence) that HB or NAG's practical performance gains are a by-product of mini-batching.

Furthermore, this work provides a viable (and provable) alternative, which, on the same set of problem instances, significantly improves over HB, NAG, and SGD's performance. This algorithm, referred to as Accelerated Stochastic Gradient Descent (ASGD), is a simple to implement stochastic algorithm, based on a relatively less popular variant of Nesterov's Acceleration. Extensive empirical results in this paper show that ASGD has performance gains over HB, NAG, and SGD. The code implementing the ASGD Algorithm can be found here[1].

## 1 Introduction

First order optimization methods, which access a function (to be optimized) through its gradient or an unbiased approximation of its gradient, are the workhorses for modern large scale optimization problems, which include training the current state-of-the-art deep neural networks. Gradient descent (Cauchy, 1847) is the simplest first order method that is used heavily in practice. However, it is known that for the class of smooth convex functions as well as some simple non-smooth problems (Nesterov, 2012a)), gradient descent is suboptimal (Nesterov, 2004) and there exists a class of algorithms called fast gradient/momentum based methods which achieve optimal convergence guarantees. The heavy ball method (Polyak, 1964) and Nesterov's accelerated gradient descent (Nesterov, 1983) are two of the most popular methods in this category.

On the other hand, training deep neural networks on large scale datasets have been possible through the use of Stochastic Gradient Descent (SGD) (Robbins & Monro, 1951), which samples a random subset of training data to compute gradient estimates that are then used to optimize the objective function. The advantages of SGD for large scale optimization and the related issues of tradeoffs between computational and statistical efficiency was highlighted in Bottou & Bousquet (2007).

---

[*]part of the work was done during an internship at Microsoft Research, India.
[1]link to the ASGD code: https://github.com/rahulkidambi/AccSGD

The above mentioned theoretical advantages of fast gradient methods (Polyak, 1964; Nesterov, 1983) (albeit for smooth convex problems) coupled with cheap to compute stochastic gradient estimates led to the influential work of Sutskever et al. (2013), which demonstrated the empirical advantages possessed by SGD when augmented with the momentum machinery. This work has led to widespread adoption of momentum methods for training deep neural nets; so much so that, in the context of neural network training, gradient descent often refers to momentum methods.

But, there is a subtle difference between classical momentum methods and their implementation in practice – classical momentum methods work in the exact first order oracle model (Nesterov, 2004), i.e., they employ *exact gradients* (computed on the full training dataset), while in practice (Sutskever et al., 2013), they are implemented with *stochastic gradients* (estimated from a randomly sampled mini-batch of training data). This leads to a natural question:

*"Are momentum methods optimal even in the* **stochastic first order oracle** *(SFO) model, where we access stochastic gradients computed on a small constant sized minibatches (or a batchsize of* $1$*?)"*

Even disregarding the question of optimality of momentum methods in the SFO model, it is not even known if momentum methods (say, Polyak (1964); Nesterov (1983)) provide *any provable* improvement over SGD in this model. While these are open questions, a recent effort of Jain et al. (2017) showed that improving upon SGD (in the stochastic first order oracle) is rather subtle as there exists problem instances in SFO model where it is not possible to improve upon SGD, even information theoretically. Jain et al. (2017) studied a variant of Nesterov's accelerated gradient updates (Nesterov, 2012b) for stochastic linear regression and show that their method improves upon SGD wherever it is information theoretically admissible. Through out this paper, we refer to the algorithm of Jain et al. (2017) as Accelerated Stochastic Gradient Method (ASGD) while we refer to a stochastic version of the most widespread form of Nesterov's method (Nesterov, 1983) as NAG; HB denotes a stochastic version of the heavy ball method (Polyak, 1964). Critically, while Jain et al. (2017) shows that ASGD improves on SGD in any information-theoretically admissible regime, it is still not known whether HB and NAG can achieve a similar performance gain.

A key contribution of this work is to show that HB *does not* provide similar performance gains over SGD even when it is informationally-theoretically admissible. That is, we provide a problem instance where it is indeed possible to improve upon SGD (and ASGD achieves this improvement), but HB *cannot* achieve any improvement over SGD. We validate this claim empirically as well. In fact, we provide empirical evidence to the claim that NAG also do not achieve any improvement over SGD for several problems where ASGD can still achieve better rates of convergence.

This raises a question about why HB and NAG provide better performance than SGD in practice (Sutskever et al., 2013), especially for training deep networks. Our conclusion (that is well supported by our theoretical result) is that HB and NAG's improved performance is attributed to mini-batching and hence, these methods will often struggle to improve over SGD with small constant batch sizes. This is in stark contrast to methods like ASGD, which is designed to improve over SGD across both small or large mini-batch sizes. In fact, based on our experiments, we observe that on the task of training deep residual networks (He et al., 2016a) on the cifar-10 dataset, we note that ASGD offers noticeable improvements by achieving $5 - 7\%$ better test error over HB and NAG even with commonly used batch sizes like $128$ during the initial stages of the optimization.

## 1.1 CONTRIBUTIONS

The contributions of this paper are as follows.

1. In Section 3, we prove that HB is *not optimal* in the SFO model. In particular, there exist linear regression problems for which the performance of HB (with *any* step size and momentum) is either the same or worse than that of SGD while ASGD improves upon both of them.

2. Experiments on several linear regression problems suggest that the suboptimality of HB in the SFO model is not restricted to special cases – it is rather widespread. Empirically, the same holds true for NAG as well (Section 5).

3. The above observations suggest that the only reason for the superiority of momentum methods in practice is mini-batching, which reduces the variance in stochastic gradients and moves the SFO closer to the exact first order oracle. This conclusion is supported by em-

| **Algorithm 1** HB: Heavy ball with a SFO | **Algorithm 2** NAG: Nesterov's AGD with a SFO |
|---|---|
| **Require:** Initial $w_0$, stepsize $\delta$, momentum $\alpha$ | **Require:** Initial $w_0$, stepsize $\delta$, momentum $\alpha$ |
| 1: $w_{-1} \leftarrow w_0; t \leftarrow 0$    /*Set $w_{-1}$ to $w_0$*/ | 1: $v_0 \leftarrow w_0; t \leftarrow 0$    /*Set $v_0$ to $w_0$*/ |
| 2: **while** $w_t$ not converged **do** | 2: **while** $w_t$ not converged **do** |
| 3:    $w_{t+1} \leftarrow w_t - \delta \cdot \widehat{\nabla} f_t(w_t) + \alpha \cdot (w_t - w_{t-1})$ /*Sum of stochastic gradient step and momentum*/ | 3:    $v_{t+1} \leftarrow w_t - \delta \cdot \widehat{\nabla} f_t(w_t)$ /*SGD step*/ |
| | 4:    $w_{t+1} = (1 + \alpha)v_{t+1} - \alpha v_t$/*Sum of SGD step and previous iterate*/ |
| 4:    $t \leftarrow t + 1$ | 5:    $t \leftarrow t + 1$ |
| **Ensure:** $w_t$    /*Return the last iterate*/ | **Ensure:** $w_t$    /*Return the last iterate*/ |

pirical evidence through training deep residual networks on cifar-10, with a batch size of $8$ (see Section 5.3).

4. We present an intuitive and easier to tune version of ASGD (see Section 4) and show that ASGD can provide significantly faster convergence to a reasonable accuracy than SGD, HB, NAG, while still providing favorable or comparable asymptotic accuracy as these methods, particularly on several deep learning problems.

Hence, the take-home message of this paper is: *HB and NAG are not optimal in the SFO model. The only reason for the superiority of momentum methods in practice is mini-batching. ASGD provides a distinct advantage in training deep networks over SGD, HB and NAG.*

## 2 NOTATION

We denote matrices by bold-face capital letters and vectors by lower-case letters. $f(w) = 1/n \sum_i f_i(w)$ denotes the function to optimize w.r.t. model parameters $w$. $\nabla f(w)$ denotes exact gradient of $f$ at $w$ while $\widehat{\nabla} f_t(w)$ denotes a stochastic gradient of $f$. That is, $\widehat{\nabla} f_t(w_t) = \nabla f_{i_t}(w)$ where $i_t$ is sampled uniformly at random from $[1, \ldots, n]$. For linear regression, $f_i(w) = 0.5 \cdot (b_i - \langle w, a_i \rangle)^2$ where $b_i \in \Re$ is the target and $a_i \in \Re^d$ is the covariate, and $\widehat{\nabla} f_t(w_t) = -(b_t - \langle w_t, a_t \rangle)a_t$. In this case, $\mathbf{H} = \mathbb{E}\left[aa^\top\right]$ denotes the Hessian of $f$ and $\kappa = \frac{\lambda_1(\mathbf{H})}{\lambda_d(\mathbf{H})}$ denotes it's condition number.

Algorithm 1 provides a pseudo-code of HB method (Polyak, 1964). $w_t - w_{t-1}$ is the momentum term and $\alpha$ denotes the momentum parameter. Next iterate $w_{t+1}$ is obtained by a linear combination of the SGD update and the momentum term. Algorithm 2 provides pseudo-code of a stochastic version of the most commonly used form of Nesterov's accelerated gradient descent (Nesterov, 1983).

## 3 SUBOPTIMALITY OF HEAVY BALL METHOD

In this section, we show that there exists linear regression problems where the performance of HB (Algorithm 1) is no better than that of SGD, while ASGD significantly improves upon SGD's performance. Let us now describe the problem instance.

Fix $w^* \in \mathbb{R}^2$ and let $(a, b) \sim \mathcal{D}$ be a sample from the distribution such that:

$$a = \begin{cases} \sigma_1 \cdot z \cdot e_1 \text{ w.p. } 0.5 \\ \sigma_2 \cdot z \cdot e_2 \text{ w.p. } 0.5, \end{cases} \quad \text{and} \quad b = \langle w^*, a \rangle,$$

where $e_1, e_2 \in \mathbb{R}^2$ are canonical basis vectors, $\sigma_1 > \sigma_2 > 0$. Let $z$ be a random variable such that $\mathbb{E}\left[z^2\right] = 2$ and $\mathbb{E}\left[z^4\right] = 2c \geq 4$. Hence, we have: $\mathbb{E}\left[(a^{(i)})^2\right] = \sigma_i^2, \mathbb{E}\left[(a^{(i)})^4\right] = c\sigma_i^4$, for $i = 1, 2$. Now, our goal is to minimize:

$$f(w) \stackrel{\text{def}}{=} 0.5 \cdot \mathbb{E}\left[(\langle w^*, a \rangle - b)^2\right], \text{ Hessian } \mathbf{H} \stackrel{\text{def}}{=} \mathbb{E}\left[aa^\top\right] = \begin{bmatrix} \sigma_1^2 & 0 \\ 0 & \sigma_2^2 \end{bmatrix}.$$

Let $\kappa$ and $\tilde{\kappa}$ denote the *computational* and *statistical* condition numbers – see Jain et al. (2017) for definitions. For the problem above, we have $\kappa = \frac{c\sigma_1^2}{\sigma_2^2}$ and $\tilde{\kappa} = c$. Then we obtain following convergence rates for SGD and ASGD when applied to the above given problem instance:

---

**Algorithm 3** Accelerated stochastic gradient descent – ASGD

---

**Input:** Initial $w_0$, short step $\delta$, long step parameter $\kappa \geq 1$, statistical advantage parameter $\xi \leq \sqrt{\kappa}$

1:   $\bar{w}_0 \leftarrow w_0$; $t \leftarrow 0$                                                 /*Set running average to $w_0$*/

2:   $\alpha \leftarrow 1 - \frac{0.7^2 \cdot \xi}{\kappa}$                                                       /*Set momentum value*/

3:   **while** $w_t$ not converged **do**

4:      $\bar{w}_{t+1} \leftarrow \alpha \cdot \bar{w}_t + (1 - \alpha) \cdot \left( w_t - \frac{\kappa \cdot \delta}{0.7} \cdot \widehat{\nabla} f_t(w_t) \right)$       /*Update the running average as a weighted average of previous running average and a long step gradient */

5:      $w_{t+1} \leftarrow \frac{0.7}{0.7+(1-\alpha)} \cdot \left( w_t - \delta \cdot \widehat{\nabla} f_t(w_t) \right) + \frac{1-\alpha}{0.7+(1-\alpha)} \cdot \bar{w}_{t+1}$       /*Update the iterate as weighted average of current running average and short step gradient*/

6:      $t \leftarrow t + 1$

**Output:**   $w_t$                                                             /*Return the last iterate*/

---

**Corollary 1** (of Theorem 1 of Jain et al. (2016)). *Let $w_t^{SGD}$ be the $t^{th}$ iterate of SGD on the above problem with starting point $w_0$ and stepsize $\frac{1}{c\sigma_1^2}$. The error of $w_t^{SGD}$ can be bounded as,*

$$\mathbb{E}\left[ f\left( w_t^{SGD} \right) \right] - f\left( w_* \right) \leq \exp\left( \frac{-t}{\kappa} \right) \left( f\left( w_0 \right) - f\left( w_* \right) \right).$$

On the other hand, ASGD achieves the following superior rate.

**Corollary 2** (of Theorem 1 of Jain et al. (2017)). *Let $w_t^{ASGD}$ be the $t^{th}$ iterate of ASGD on the above problem with starting point $w_0$ and appropriate parameters. The error of $w_t^{ASGD}$ can be bounded as,*

$$\mathbb{E}\left[ f\left( w_t^{ASGD} \right) \right] - f\left( w_* \right) \leq \mathrm{poly}(\kappa) \exp\left( \frac{-t}{\sqrt{\kappa\tilde{\kappa}}} \right) \left( f\left( w_0 \right) - f\left( w_* \right) \right).$$

Note that for a given problem/input distribution $\tilde{\kappa} = c$ is a constant while $\kappa = \frac{c\sigma_1^2}{\sigma_2^2}$ can be arbitrarily large. Note that $\kappa > \tilde{\kappa} = c$. Hence, ASGD improves upon rate of SGD by a factor of $\sqrt{\kappa}$. The following proposition, which is the main result of this section, establishes that HB (Algorithm 1) cannot provide a similar improvement over SGD as what ASGD offers. In fact, we show no matter the choice of parameters of HB, its performance does not improve over SGD by more than a constant.

**Proposition 3.** *Let $w_t^{HB}$ be the $t^{th}$ iterate of HB (Algorithm 1) on the above problem with starting point $w_0$. For any choice of stepsize $\delta$ and momentum $\alpha \in [0,1]$, $\exists T$ large enough such that $\forall t \geq T$, we have,*

$$\mathbb{E}\left[ f\left( w_t^{HB} \right) \right] - f\left( w_* \right) \geq C(\kappa, \delta, \alpha) \cdot \exp\left( \frac{-500t}{\kappa} \right) \left( f\left( w_0 \right) - f\left( w_* \right) \right),$$

*where $C(\kappa, \delta, \alpha)$ depends on $\kappa, \delta$ and $\alpha$ (but not on $t$).*

Thus, to obtain $\widehat{w}$ s.t. $\|\widehat{w} - w^*\| \leq \epsilon$, HB requires $\Omega(\kappa \log \frac{1}{\epsilon})$ samples and iterations. On the other hand, ASGD can obtain $\epsilon$-approximation to $w^*$ in $\mathcal{O}(\sqrt{\kappa} \log \kappa \log \frac{1}{\epsilon})$ iterations. We note that the gains offered by ASGD are meaningful when $\kappa > \mathcal{O}(c)$ (Jain et al., 2017); otherwise, all the algorithms including SGD achieve nearly the same rates (upto constant factors). While we do not prove it theoretically, we observe empirically that for the same problem instance, NAG also obtains nearly same rate as HB and SGD. We conjecture that a lower bound for NAG can be established using a similar proof technique as that of HB (i.e. Proposition 3). We also believe that the constant in the lower bound described in proposition 3 can be improved to some small number ($\leq 5$).

## 4   ALGORITHM

We will now present and explain an intuitive version of ASGD (pseudo code in Algorithm 3). The algorithm takes three inputs: short step $\delta$, long step parameter $\kappa$ and statistical advantage parameter $\xi$. The short step $\delta$ is precisely the same as the step size in SGD, HB or NAG. For convex problems, this scales inversely with the smoothness of the function. The long step parameter $\kappa$ is intended

to give an estimate of the ratio of the largest and smallest curvatures of the function; for convex functions, this is just the condition number. The statistical advantage parameter $\xi$ captures trade off between statistical and computational condition numbers – in the deterministic case, $\xi = \sqrt{\kappa}$ and ASGD is equivalent to NAG, while in the high stochasticity regime, $\xi$ is much smaller. The algorithm maintains two iterates: descent iterate $w_t$ and a running average $\bar{w}_t$. The running average is a weighted average of the previous average and a long gradient step from the descent iterate, while the descent iterate is updated as a convex combination of short gradient step from the descent iterate and the running average. The idea is that since the algorithm takes a long step as well as short step and an appropriate average of both of them, it can make progress on different directions at a similar pace. Appendix B shows the equivalence between Algorithm 3 and ASGD as proposed in Jain et al. (2017). Note that the constant 0.7 appearing in Algorithm 3 has no special significance. Jain et al. (2017) require it to be smaller than $\sqrt{1/6}$ but any constant smaller than 1 seems to work in practice.

## 5 EXPERIMENTS

We now present our experimental results exploring performance of SGD, HB, NAG and ASGD. Our experiments are geared towards answering the following questions:

- Even for linear regression, is the suboptimality of HB restricted to specific distributions in Section 3 or does it hold for more general distributions as well? Is the same true of NAG?
- What is the reason for the superiority of HB and NAG in practice? Is it because momentum methods have better performance that SGD for stochastic gradients or due to minibatching? Does this superiority hold even for small minibatches?
- How does the performance of ASGD compare to that of SGD, HB and NAG, when training deep networks?

Section 5.1 and parts of Section 5.2 address the first two questions. Section 5.2 and 5.3 address Question 2 partially and the last question. We use Matlab to conduct experiments presented in Section 5.1 and use PyTorch (pytorch, 2017) for our deep networks related experiments. Pytorch code implementing the ASGD algorithm can be found at https://github.com/rahulkidambi/AccSGD.

### 5.1 LINEAR REGRESSION

In this section, we will present results on performance of the four optimization methods (SGD, HB, NAG, and ASGD) for linear regression problems. We consider two different class of linear regression problems, both of them in two dimensions. Given $\kappa$ which stands for condition number, we consider the following two distributions:

**Discrete**: $a = e_1$ w.p. 0.5 and $a = \frac{2}{\kappa} \cdot e_2$ with 0.5; $e_i$ is the $i^{th}$ standard basis vector.

**Gaussian** : $a \in \mathbb{R}^2$ is distributed as a Gaussian random vector with covariance matrix $\begin{bmatrix} 1 & 0 \\ 0 & \frac{1}{\kappa} \end{bmatrix}$.

We fix a randomly generated $w^* \in \mathbb{R}^2$ and for both the distributions above, we let $b = \langle w^*, a \rangle$. We vary $\kappa$ from $\{2^4, 2^5, ..., 2^{12}\}$ and for each $\kappa$ in this set, we run 100 independent runs of all four methods, each for a total of $t = 5\kappa$ iterations. We define that the algorithm converges if there is no error in the second half (i.e. after $2.5\kappa$ updates) that exceeds the starting error - this is reasonable since we expect *geometric convergence* of the initial error.

Unlike ASGD and SGD, we do not know optimal learning rate and momentum parameters for NAG and HB in the stochastic gradient model. So, we perform a *grid search* over the values of the learning rate and momentum parameters. In particular, we lay a $10 \times 10$ grid in $[0, 1] \times [0, 1]$ for learning rate and momentum and run NAG and HB. Then, for each grid point, we consider the subset of 100 trials that converged and computed the final error using these. Finally, the parameters that yield the minimal error are chosen for NAG and HB, and these numbers are reported. We measure convergence performance of a method using:

$$\text{rate} = \frac{\log(f(w_0)) - \log(f(w_t))}{t}, \tag{1}$$

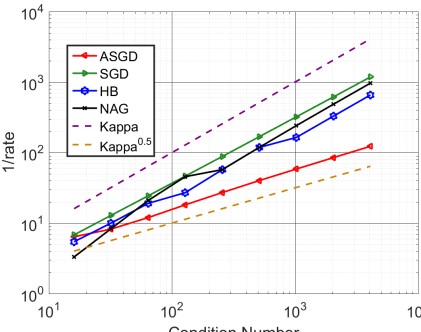 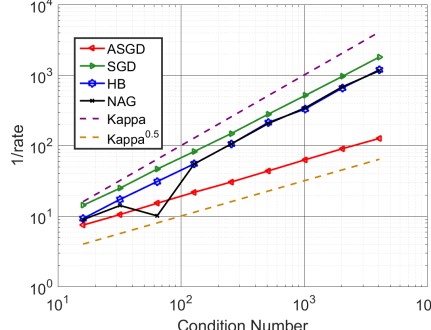

Figure 1: Plot of 1/rate (refer equation (1)) vs condition number ($\kappa$) for various methods for the linear regression problem. Discrete distribution in the left, Gaussian to the right.

| Algorithm | Slope – discrete | Slope – Gaussian |
|-----------|------------------|------------------|
| SGD | 0.9302 | 0.8745 |
| HB | 0.8522 | 0.8769 |
| NAG | 0.98 | 0.9494 |
| ASGD | 0.5480 | 0.5127 |

Table 1: Slopes (i.e. $\gamma$) obtained by fitting a line to the curves in Figure 1. A value of $\gamma$ indicates that the error decays at a rate of $\exp\left(\frac{-t}{\kappa^\gamma}\right)$. A smaller value of $\gamma$ indicates a faster rate of error decay.

We compute the rate (1) for all the algorithms with varying condition number $\kappa$. Given a rate vs $\kappa$ plot for a method, we compute it's *slope* (denoted as $\gamma$) using linear regression. Table 1 presents the estimated slopes (i.e. $\gamma$) for various methods for both the discrete and the Gaussian case. The slope values clearly show that the rate of SGD, HB and NAG have a nearly linear dependence on $\kappa$ while that of ASGD seems to scale linearly with $\sqrt{\kappa}$.

## 5.2 DEEP AUTOENCODERS FOR MNIST

In this section, we present experimental results on training deep autoencoders for the mnist dataset, and we closely follow the setup of Hinton & Salakhutdinov (2006). This problem is a standard benchmark for evaluating the performance of different optimization algorithms e.g., Martens (2010); Sutskever et al. (2013); Martens & Grosse (2015); Reddi et al. (2017). The network architecture follows previous work (Hinton & Salakhutdinov, 2006) and is represented as $784 - 1000 - 500 - 250 - 30 - 250 - 500 - 1000 - 784$ with the first and last $784$ nodes representing the input and output respectively. All hidden/output nodes employ sigmoid activations except for the layer with 30 nodes which employs linear activations and we use MSE loss. Initialization follows the scheme of Martens (2010), also employed in Sutskever et al. (2013); Martens & Grosse (2015). We perform training with two minibatch sizes $-1$ and 8. The runs with minibatch size of 1 were run for 30 epochs while the runs with minibatch size of 8 were run for 50 epochs. For each of SGD, HB, NAG and ASGD, a grid search over learning rate, momentum and long step parameter (whichever is applicable) was done and best parameters were chosen based on achieving the smallest training error in the same protocol followed by Sutskever et al. (2013). The grid was extended whenever the best parameter fell at the edge of a grid. For the parameters chosen by grid search, we perform 10 runs with different seeds and averaged the results. The results are presented in Figures 2 and 3. Note that the final loss values reported are suboptimal compared to those in published literature e.g., Sutskever et al. (2013); while Sutskever et al. (2013) report results after $750000$ updates with a large batch size of 200 (which implies a total of $750000 \times 200 = 150M$ gradient evaluations), whereas, our results are after 1.8M updates of SGD with a batch size 1 (which is just 1.8M gradient evaluations).

**Effect of minibatch sizes**: While HB and NAG decay the loss faster compared to SGD for a minibatch size of 8 (Figure 2), this superior decay rate does not hold for a minibatch size of 1 (Figure 3). This supports our intuitions from the stochastic linear regression setting, where we demonstrate that HB and NAG are suboptimal in the stochastic first order oracle model.

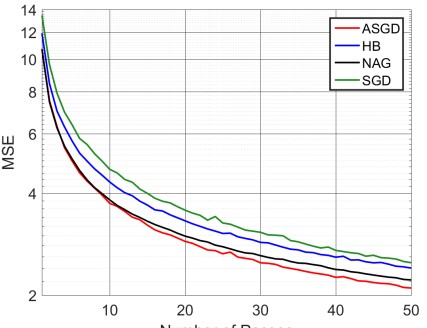 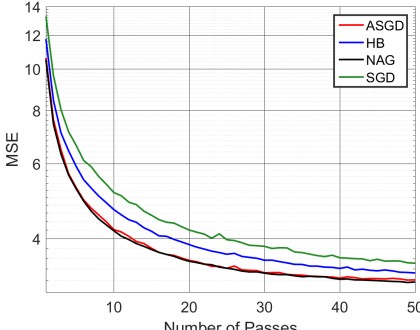

Figure 2: Training loss (left) and test loss (right) while training deep autoencoder for mnist with minibatch size 8. Clearly, ASGD matches performance of NAG and outperforms SGD on the test data. HB also outperforms SGD.

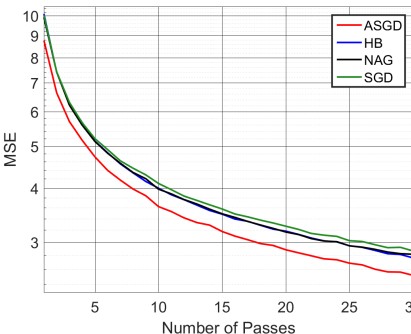 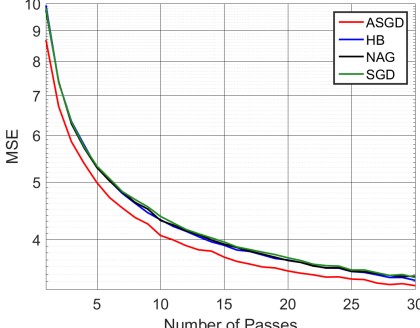

Figure 3: Training loss (left) and test loss (right) while training deep autoencoder for mnist with minibatch size 1. Interestingly, SGD, HB and NAG, all decrease the loss at a similar rate, while ASGD decays at a faster rate.

**Comparison of ASGD with momentum methods**: While ASGD performs slightly better than NAG for batch size 8 in the training error (Figure 2), ASGD decays the error at a faster rate compared to all the three other methods for a batch size of 1 (Figure 3).

## 5.3    DEEP RESIDUAL NETWORKS FOR CIFAR-10

We will now present experimental results on training deep residual networks (He et al., 2016b) with pre-activation blocks He et al. (2016a) for classifying images in cifar-10 (Krizhevsky & Hinton, 2009); the network we use has 44 layers (dubbed preresnet-44). The code for this section was downloaded from preresnet (2017). One of the most distinct characteristics of this experiment compared to our previous experiments is learning rate decay. We use a validation set based decay scheme, wherein, after every 3 epochs, we decay the learning rate by a certain factor (which we grid search on) if the validation zero one error does not decrease by at least a certain amount (precise numbers are provided in the appendix since they vary across batch sizes). Due to space constraints, we present only a subset of training error plots. Please see Appendix C.3 for some more plots on training errors.

**Effect of minibatch sizes**: Our first experiment tries to understand how the performance of HB and NAG compare with that of SGD and how it varies with minibatch sizes. Figure 4 presents the test zero one error for minibatch sizes of 8 and 128. While training with batch size 8 was done for 40 epochs, with batch size 128, it was done for 120 epochs. We perform a grid search over all parameters for each of these algorithms. See Appendix C.3 for details on the grid search parameters. We observe that final error achieved by SGD, HB and NAG are all very close for both batch sizes. While NAG exhibits a superior rate of convergence compared to SGD and HB for batch size 128, this superior rate of convergence disappears for a batch size of 8.

**Comparison of ASGD with momentum methods**: The next experiment tries to understand how ASGD compares with HB and NAG. The errors achieved by various methods when we do

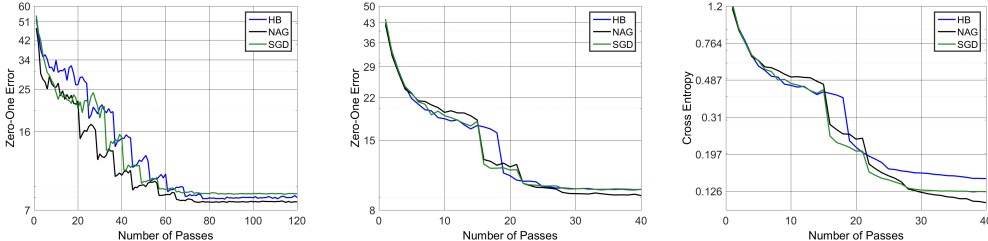

Figure 4: Test zero one loss for batch size $128$ (left), batch size $8$ (center) and training function value for batch size $8$ (right) for SGD, HB and NAG.

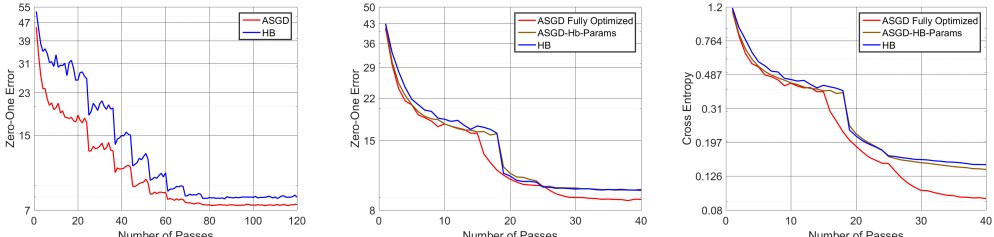

Figure 5: Test zero one loss for batch size $128$ (left), batch size $8$ (center) and training function value for batch size $8$ (right) for ASGD compared to HB. In the above plots, both ASGD and ASGD-Hb-Params refer to ASGD run with the learning rate and decay schedule of HB. ASGD-Fully-Optimized refers to ASGD where learning rate and decay schedule were also selected by grid search.

grid search over all parameters are presented in Table 2. Note that the final test errors for batch size $128$ are better than those for batch size $8$ since the former was run for $120$ epochs while the latter was run only for $40$ epochs (due to time constraints).

| Algorithm | Final test error – batch size $128$ | Final test error – batch size $8$ |
|---|---|---|
| SGD | $8.32 \pm 0.21$ | $9.57 \pm 0.18$ |
| HB | $7.98 \pm 0.19$ | $9.28 \pm 0.25$ |
| NAG | $7.63 \pm 0.18$ | $9.07 \pm 0.18$ |
| ASGD | $\mathbf{7.23 \pm 0.22}$ | $\mathbf{8.52 \pm 0.16}$ |

Table 2: Final test errors achieved by various methods for batch sizes of $128$ and $8$. The hyperparameters have been chosen by grid search.

While the final error achieved by ASGD is similar/favorable compared to all other methods, we are also interested in understanding whether ASGD has a superior convergence speed. For this experiment, we need to address the issue of differing learning rates used by various algorithms and different iterations where they decay learning rates. So, for each of HB and NAG, we choose the learning rate and decay factors by grid search, use these values for ASGD and do grid search only over long step parameter $\kappa$ and momentum $\alpha$ for ASGD. The results are presented in Figures 5 and 6. For batch size $128$, ASGD decays error at a faster rate compared to both HB and NAG. For batch size $8$, while we see a superior convergence of ASGD compared to NAG, we do not see this superiority over HB. The reason for this turns out to be that the learning rate for HB, which we also use for ASGD, turns out to be quite suboptimal for ASGD. So, for batch size $8$, we also compare fully optimized (i.e., grid search over learning rate as well) ASGD with HB. The superiority of ASGD over HB is clear from this comparison. These results suggest that ASGD decays error at a faster rate compared to HB and NAG across different batch sizes.

# 6 RELATED WORK

**First order oracle methods**: The primary method in this family is Gradient Descent (GD) (Cauchy, 1847). As mentioned previously, GD is suboptimal for smooth convex optimization (Nesterov,

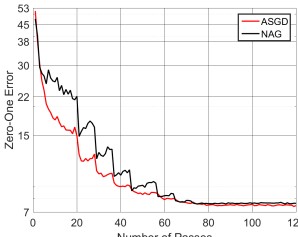 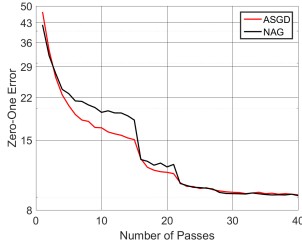 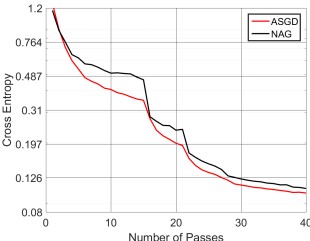

Figure 6: Test zero one loss for batch size 128 (left), batch size 8 (center) and training function value for batch size 8 (right) for ASGD compared to NAG. In the above plots, ASGD was run with the learning rate and decay schedule of NAG. Other parameters were selected by grid search.

2004), and this is addressed using momentum methods such as the Heavy Ball method (Polyak, 1964) (for quadratics), and Nesterov's Accelerated gradient descent (Nesterov, 1983).

**Stochastic first order methods and noise stability**: The simplest method employing the SFO is SGD (Robbins & Monro, 1951); the effectiveness of SGD has been immense, and its applicability goes well beyond optimizing convex objectives. Accelerating SGD is a tricky proposition given the instability of fast gradient methods in dealing with noise, as evidenced by several negative results which consider statistical (Proakis, 1974; Polyak, 1987; Roy & Shynk, 1990), numerical (Paige, 1971; Greenbaum, 1989) and adversarial errors (d'Aspremont, 2008; Devolder et al., 2014). A result of Jain et al. (2017) developed the first provably accelerated SGD method for linear regression which achieved minimax rates, inspired by a method of Nesterov (2012b). Schemes of Ghadimi & Lan (2012; 2013); Dieuleveut et al. (2016), which indicate acceleration is possible with noisy gradients do not hold in the SFO model satisfied by algorithms that are run in practice (see Jain et al. (2017) for more details).

While HB (Polyak, 1964) and NAG (Nesterov, 1983) are known to be effective in case of exact first order oracle, for the SFO, the theoretical performance of HB and NAG is not well understood.

**Understanding Stochastic Heavy Ball:** Understanding HB's performance with inexact gradients has been considered in efforts spanning several decades, in many communities like controls, optimization and signal processing. Polyak (1987) considered HB with noisy gradients and concluded that the improvements offered by HB with inexact gradients vanish unless strong assumptions on the inexactness was considered; an instance of this is when the variance of inexactness decreased as the iterates approach the minimizer. Proakis (1974); Roy & Shynk (1990); Sharma et al. (1998) suggest that the improved non-asymptotic rates offered by stochastic HB arose at the cost of worse asymptotic behavior. We resolve these unquantified improvements on rates as being just constant factors over SGD, in stark contrast to the gains offered by ASGD. Loizou & Richtárik (2017) state their method as Stochastic HB but require stochastic gradients that nearly behave as exact gradients; indeed, their rates match that of the standard HB method (Polyak, 1964). Such rates are not information theoretically possible (see Jain et al. (2017)), especially with a batch size of 1 or even with constant sized minibatches.

**Accelerated and Fast Methods for finite-sums:** There have been developments pertaining to faster methods for finite-sums (also known as offline stochastic optimization): amongst these are methods such as SDCA (Shalev-Shwartz & Zhang, 2012), SAG (Roux et al., 2012), SVRG (Johnson & Zhang, 2013), SAGA (Defazio et al., 2014), which offer linear convergence rates for strongly convex finite-sums, improving over SGD's sub-linear rates (Rakhlin et al., 2012). These methods have been improved using accelerated variants (Shalev-Shwartz & Zhang, 2014; Frostig et al., 2015a; Lin et al., 2015; Defazio, 2016; Allen-Zhu, 2016). Note that these methods require storing the entire training set in memory and taking multiple passes over the same for guaranteed progress. Furthermore, these methods require computing a batch gradient or require memory requirements (typically $\Omega(|$ training data points$|)$). For deep learning problems, data augmentation is often deemed necessary for achieving good performance; this implies computing quantities such as batch gradient (or storage necessities) over this augmented dataset is often infeasible. Such requirements are mitigated by the use of simple streaming methods such as SGD, ASGD, HB, NAG. For other technical distinctions between the offline and online stochastic methods refer to Frostig et al. (2015b).

**Practical methods for training deep networks**: Momentum based methods employed with stochastic gradients (Sutskever et al., 2013) have become standard and very popular in practice. These schemes tend to outperform standard SGD on several important practical problems. As previously mentioned, we attribute this improvement to effect of mini-batching rather than improvement offered by HB or NAG in the SFO model. Schemes such as Adagrad (Duchi et al., 2011), RMSProp (Tieleman & Hinton, 2012), Adam (Kingma & Ba, 2014) represent an important and useful class of algorithms. The advantages offered by these methods are orthogonal to the advantages offered by fast gradient methods; it is an important direction to explore augmenting these methods with ASGD as opposed to standard HB or NAG based acceleration schemes.

Chaudhari et al. (2017) proposed Entropy-SGD, which is an altered objective that adds a local strong convexity term to the actual empirical risk objective, with an aim to improve generalization. However, we do not understand convergence rates for convex problems or the generalization ability of this technique in a rigorous manner. Chaudhari et al. (2017) propose to use SGD in their procedure but mention that they employ the HB/NAG method in their implementation for achieving better performance. Naturally, we can use ASGD in this context. Path normalized SGD (Neyshabur et al., 2015) is a variant of SGD that alters the metric on which the weights are optimized. As noted in their paper, path normalized SGD could be improved using HB/NAG (or even the ASGD method).

## 7    CONCLUSIONS AND FUTURE DIRECTIONS

In this paper, we show that the performance gain of HB over SGD in stochastic setting is attributed to mini-batching rather than the algorithm's ability to *accelerate* with stochastic gradients. Concretely, we provide a formal proof that for several *easy* problem instances, HB does not outperform SGD despite large condition number of the problem; we observe this trend for NAG in our experiments. In contrast, ASGD (Jain et al., 2017) provides significant improvement over SGD for these problem instances. We observe similar trends when training a resnet on cifar-10 and an autoencoder on mnist. This work motivates several directions such as understanding the behavior of ASGD on domains such as NLP, and developing automatic momentum tuning schemes (Zhang et al., 2017).

ACKNOWLEDGMENTS

Sham Kakade acknowledges funding from NSF Awards CCF-1703574 and CCF-1740551.

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

# A   SUBOPTIMALITY OF HB: PROOF OF PROPOSITION 3

Before proceeding to the proof, we introduce some additional notation. Let $\boldsymbol{\theta}_{t+1}^{(j)}$ denote the concatenated and centered estimates in the $j^{\text{th}}$ direction for $j = 1, 2$.

$$\boldsymbol{\theta}_{t+1}^{(j)} \overset{\text{def}}{=} \begin{bmatrix} \mathbf{w}_{t+1}^{(j)} - (\mathbf{w}^*)^{(j)} \\ \mathbf{w}_t^{(j)} - (\mathbf{w}^*)^{(j)} \end{bmatrix}, \quad j = 1, 2.$$

Since the distribution over $x$ is such that the coordinates are decoupled, we see that $\boldsymbol{\theta}_{t+1}^{(j)}$ can be written in terms of $\boldsymbol{\theta}_t^{(j)}$ as:

$$\boldsymbol{\theta}_{t+1}^{(j)} = \widehat{\mathbf{A}}_{t+1}^{(j)} \boldsymbol{\theta}_t^{(j)}, \text{ with } \widehat{\mathbf{A}}_{t+1}^{(j)} = \begin{bmatrix} 1 + \alpha - \delta(a_{t+1}^{(j)})^2 & -\alpha \\ 1 & 0 \end{bmatrix}.$$

Let $\boldsymbol{\Phi}_{t+1}^{(j)} \overset{\text{def}}{=} \mathbb{E} \left[ \boldsymbol{\theta}_{t+1}^{(j)} \otimes \boldsymbol{\theta}_{t+1}^{(j)} \right]$ denote the covariance matrix of $\boldsymbol{\theta}_{t+1}^{(j)}$. We have $\boldsymbol{\Phi}_{t+1}^{(j)} = \mathcal{B}^{(j)} \boldsymbol{\Phi}_t^{(j)}$ with, $\mathcal{B}^{(j)}$ defined as

$$\mathcal{B}^{(j)} \overset{\text{def}}{=} \begin{bmatrix} \mathbb{E}\left[(1 + \alpha - \delta(a^{(j)})^2)^2\right] & \mathbb{E}\left[-\alpha(1 + \alpha - \delta(a^{(j)})^2)\right] & \mathbb{E}\left[-\alpha(1 + \alpha - \delta(a^{(j)})^2\right] & \alpha^2 \\ \mathbb{E}\left[(1 + \alpha - \delta(a^{(j)})^2)\right] & 0 & -\alpha & 0 \\ \mathbb{E}\left[(1 + \alpha - \delta(a^{(j)})^2)\right] & -\alpha & 0 & 0 \\ 1 & 0 & 0 & 0 \end{bmatrix}$$

$$= \begin{bmatrix} (1 + \alpha - \delta\sigma_j^2)^2 + (c - 1)(\delta\sigma_j^2)^2 & -\alpha(1 + \alpha - \delta\sigma_j^2) & -\alpha(1 + \alpha - \delta\sigma_j^2) & \alpha^2 \\ (1 + \alpha - \delta\sigma_j^2) & 0 & -\alpha & 0 \\ (1 + \alpha - \delta\sigma_j^2) & -\alpha & 0 & 0 \\ 1 & 0 & 0 & 0 \end{bmatrix}.$$

We prove Proposition 3 by showing that for any choice of stepsize and momentum, either of the two holds:

- $\mathcal{B}^{(1)}$ has an eigenvalue larger than 1, or,
- the largest eigenvalue of $\mathcal{B}^{(2)}$ is greater than $1 - \frac{500}{\kappa}$.

This is formalized in the following two lemmas.

**Lemma 4.** *If the stepsize $\delta$ is such that $\delta\sigma_1^2 \geq \frac{2(1 - \alpha^2)}{c + (c-2)\alpha}$, then $\mathcal{B}^{(1)}$ has an eigenvalue $\geq 1$.*

**Lemma 5.** *If the stepsize $\delta$ is such that $\delta\sigma_1^2 < \frac{2(1 - \alpha^2)}{c + (c-2)\alpha}$, then $\mathcal{B}^{(2)}$ has an eigenvalue of magnitude $\geq 1 - \frac{500}{\kappa}$.*

Given this notation, we can now consider the $j^{th}$ dimension without the superscripts; when needed, they will be made clear in the exposition. Denoting $x \overset{\text{def}}{=} \delta\sigma^2$ and $t \overset{\text{def}}{=} 1 + \alpha - x$, we have:

$$\mathcal{B} = \begin{bmatrix} t^2 + (c - 1)x^2 & -\alpha t & -\alpha t & \alpha^2 \\ t & 0 & -\alpha & 0 \\ t & -\alpha & 0 & 0 \\ 1 & 0 & 0 & 0 \end{bmatrix}$$

## A.1   PROOF

The analysis goes via computation of the characteristic polynomial of $\mathcal{B}$ and evaluating it at different values to obtain bounds on its roots.

**Lemma 6.** *The characteristic polynomial of $\mathcal{B}$ is:*

$$D(z) = z^4 - (t^2 + (c - 1)x^2)z^3 + (2\alpha t^2 - 2\alpha^2)z^2 + (-t^2 + (c - 1)x^2)\alpha^2 z + \alpha^4.$$

*Proof.* We first begin by writing out the expression for the determinant:

$$Det(\mathcal{B} - z\mathcal{I}) = \begin{vmatrix} t^2 + (c-1)x^2 - z & -\alpha t & -\alpha t & \alpha^2 \\ t & -z & -\alpha & 0 \\ t & -\alpha & -z & 0 \\ 1 & 0 & 0 & -z \end{vmatrix}.$$

expanding along the first column, we have:

$$Det(\mathcal{B} - z\mathcal{I}) = (t^2 + (c-1)x^2 - z)(\alpha^2 z - z^3) - t(-\alpha t z^2 + \alpha^2 t z) + t(-\alpha t(\alpha z) + z \cdot \alpha t z) - (z \cdot \alpha^2 z - \alpha^4)$$
$$= (t^2 + (c-1)x^2 - z)(\alpha^2 z - z^3) - 2t(\alpha^2 t z - \alpha t z^2) - (\alpha^2 z^2 - \alpha^4).$$

Expanding the terms yields the expression in the lemma. $\square$

The next corollary follows by some simple arithmetic manipulations.

**Corollary 7.** *Substituting $z = 1 - \tau$ in the characteristic equation of Lemma 6, we have:*

$$\begin{aligned}
D(1-\tau) &= \tau^4 + \tau^3(-4 + t^2 + (c-1)x^2) + \tau^2(6 - 3t^2 - 3(c-1)x^2 - 2\alpha^2 + 2\alpha t^2) \\
&\quad + \tau(-4 + 3t^2 + 3(c-1)x^2 + 4\alpha^2 - 4\alpha t^2 - (c-1)x^2\alpha^2 + t^2\alpha^2) \\
&\quad + (1 - t^2 - (c-1)x^2 - 2\alpha^2 + 2\alpha t^2 + (c-1)x^2\alpha^2 - t^2\alpha^2 + \alpha^4) \\
&= \tau^4 + \tau^3[-(3+\alpha)(1-\alpha) - 2x(1+\alpha) + cx^2] \\
&\quad + \tau^2[(3 - 4\alpha - \alpha^2 + 2\alpha^3) - 2x(1+\alpha)(2\alpha - 3) + x^2(2\alpha - 3c)] \\
&\quad + \tau[-(1-\alpha)^2(1-\alpha^2) - 2x(3-\alpha)(1-\alpha^2) + x^2(3c - 4\alpha + (2-c)\alpha^2)] \\
&\quad + x(1-\alpha)[2(1-\alpha^2) - x(c + (c-2)\alpha)]. \qquad (2)
\end{aligned}$$

*Proof of Lemma 4.* The first observation necessary to prove the lemma is that the characteristic polynomial $D(z)$ approaches $\infty$ as $z \to \infty$, i.e., $\lim_{z \to \infty} D(z) = +\infty$.

Next, we evaluate the characteristic polynomial at 1, i.e. compute $D(1)$. This follows in a straightforward manner from corollary (7) by substituting $\tau = 0$ in equation (2), and this yields,

$$D(1) = (1-\alpha)x \cdot \left( 2(1-\alpha^2) - x(1-\alpha) - (c-1)x(1+\alpha) \right).$$

As $\alpha < 1$, $x = \delta\sigma^2 > 0$, we have the following by setting $D(1) \le 0$ and solving for $x$:

$$x \ge \frac{2(1-\alpha^2)}{c + (c-2)\alpha}.$$

Since $D(1) \le 0$ and $D(z) \ge 0$ as $z \to \infty$, there exists a root of $D(\cdot)$ which is $\ge 1$. $\square$

**Remark 8.** *The above characterization is striking in the sense that for any $c > 1$, increasing the momentum parameter $\alpha$ naturally requires the reduction in the step size $\delta$ to permit the convergence of the algorithm, which is not observed when fast gradient methods are employed in deterministic optimization. For instance, in the case of deterministic optimization, setting $c = 1$ yields $\delta\sigma_1^2 < 2(1+\alpha)$. On the other hand, when employing the stochastic heavy ball method with $x^{(j)} = 2\sigma_j^2$, we have the condition that $c = 2$, and this implies, $\delta\sigma_1^2 < \frac{2(1-\alpha^2)}{2} = 1 - \alpha^2$.*

We now prove Lemma 5. We first consider the large momentum setting.

**Lemma 9.** *When the momentum parameter $\alpha$ is set such that $1 - 450/\kappa \le \alpha \le 1$, $\mathcal{B}$ has an eigenvalue of magnitude $\ge 1 - \frac{450}{\kappa}$.*

*Proof.* This follows easily from the fact that $\det(\mathcal{B}) = \alpha^4 = \prod_{j=1}^{4} \lambda_j(\mathcal{B}) \le (\lambda_{\max}(\mathcal{B}))^4$, thus implying $1 - 450/\kappa \le \alpha \le |\lambda_{\max}(\mathcal{B})|$. $\square$

**Remark 10.** *Note that the above lemma holds for any value of the learning rate $\delta$, and holds for every eigen direction of $\mathbf{H}$. Thus, for "large" values of momentum, the behavior of stochastic heavy ball does degenerate to the behavior of stochastic gradient descent.*

We now consider the setting where momentum is bounded away from 1.

**Corollary 11.** *Consider $\mathcal{B}^{(2)}$, by substituting $\tau = l/\kappa$, $x = \delta\lambda_{\min} = c(\delta\sigma_1^2)/\kappa$ in equation (2) and accumulating terms in varying powers of $1/\kappa$, we obtain:*

$$
\begin{aligned}
G(l) \overset{def}{=}\ & \frac{c^3(\delta\sigma_1^2)^2 l^3}{\kappa^5} + \frac{l^4 - 2c(\delta\sigma_1^2)l^3(1+\alpha) + (2\alpha - 3c)c^2(\delta\sigma_1^2)^2 l^2}{\kappa^4} \\
& + \frac{-(3+\alpha)(1-\alpha)l^3 - 2(1+\alpha)(2\alpha-3)c(\delta\sigma_1^2)l^2 + (3c - 4\alpha + (2-c)\alpha^2)c^2(\delta\sigma_1^2)^2 l}{\kappa^3} \\
& + \frac{(3 - 4\alpha - \alpha^2 + 2\alpha^3)l^2 - 2c(\delta\sigma_1^2)l(3-\alpha)(1-\alpha^2) - c^2(\delta\sigma_1^2)^2(1-\alpha)(c + (c-2)\alpha)}{\kappa^2} \\
& + \frac{-(1-\alpha)^2(1-\alpha^2)l + 2c(\delta\sigma_1^2)(1-\alpha)(1-\alpha^2)}{\kappa}
\end{aligned}
\tag{3}
$$

**Lemma 12.** *Let $2 < c < 3000$, $0 \le \alpha \le 1 - \frac{450}{\kappa}$, $l = 1 + \frac{2c(\delta\sigma_1^2)}{1-\alpha}$. Then, $G(l) \le 0$.*

*Proof.* Since $(\delta\sigma_1^2) \le \frac{2(1-\alpha^2)}{c + (c-2)\alpha}$, this implies $\frac{(\delta\sigma_1^2)}{1-\alpha} \le \frac{2(1+\alpha)}{c+(c-2)\alpha} \le \frac{4}{c}$, thus implying, $1 \le l \le 9$.

Substituting the value of $l$ in equation (3), the coefficient of $\mathcal{O}(1/\kappa)$ is $-(1-\alpha)^3(1+\alpha)$.

We will bound this term along with $(3 - 4\alpha - \alpha^2 + 2\alpha^3)l^2/\kappa^2 = (1-\alpha)^2(3+2\alpha)l^2/\kappa^2$ to obtain:

$$
\begin{aligned}
\frac{-(1-\alpha)^3(1+\alpha)}{\kappa} + \frac{(1-\alpha)^2(3+2\alpha)l^2}{\kappa^2} &\le \frac{-(1-\alpha)^3(1+\alpha)}{\kappa} + \frac{405(1-\alpha)^2}{\kappa^2} \\
&\le \frac{(1-\alpha)^2}{\kappa}\left(\frac{405}{\kappa} - (1-\alpha^2)\right) \\
&\le \frac{(1-\alpha)^2}{\kappa}\left(\frac{405}{\kappa} - (1-\alpha)\right) \le -\frac{45 \cdot 450^2}{\kappa^4},
\end{aligned}
$$

where, we use the fact that $\alpha < 1$, $l \le 9$. The natural implication of this bound is that the terms that are lower order, such as $\mathcal{O}(1/\kappa^4)$ and $\mathcal{O}(1/\kappa^5)$ will be negative owing to the large constant above. Let us verify that this is indeed the case by considering the terms having powers of $\mathcal{O}(1/\kappa^4)$ and $\mathcal{O}(1/\kappa^5)$ from equation (3):

$$
\begin{aligned}
& \frac{c^3(\delta\sigma_1^2)^2 l^3}{\kappa^5} + \frac{l^4 - 2c(\delta\sigma_1^2)l^3(1+\alpha) + (2\alpha - 3c)c^2(\delta\sigma_1^2)^2 l^2}{\kappa^4} - \frac{45 \cdot 450^2}{\kappa^4} \\
\le\ & \frac{c^3(\delta\sigma_1^2)^2 l^3}{\kappa^5} + \frac{l^4}{\kappa^4} - \frac{45 \cdot 450^2}{\kappa^4} \\
\le\ & \frac{cl^3}{\kappa^5} + \frac{(9^4 - (45 \cdot 450^2))}{\kappa^4} \le \frac{9^3 c + 9^4 - (45 \cdot 450^2)}{\kappa^4}
\end{aligned}
$$

The expression above evaluates to $\le 0$ given an upperbound on the value of $c$. The expression above follows from the fact that $l \le 9$, $\kappa \ge 1$.

Next, consider the terms involving $\mathcal{O}(1/\kappa^3)$ and $\mathcal{O}(1/\kappa^2)$, in particular,

$$
\begin{aligned}
& \frac{(3c - 4\alpha + (2-c)\alpha^2)c^2(\delta\sigma_1^2)^2 l}{\kappa^3} - \frac{c^2(\delta\sigma_1^2)^2(1-\alpha)(c + (c-2)\alpha)}{\kappa^2} \\
\le\ & \frac{c^2(\delta\sigma_1^2)^2}{\kappa^2}\left(\frac{l(3c+2)}{\kappa} - (1-\alpha)(c + (c-2)\alpha)\right) \\
\le\ & \frac{c^2(\delta\sigma_1^2)^2}{\kappa^2}\left(\frac{5cl}{\kappa} - (1-\alpha)(c + (c-2)\alpha)\right) \\
\le\ & \frac{c^2(\delta\sigma_1^2)^2}{\kappa^2}\left(\frac{5cl}{\kappa} - (1-\alpha)c\right) \\
\le\ & \frac{c^3(\delta\sigma_1^2)^2}{\kappa^2}\left(\frac{5l}{\kappa} - \frac{450}{\kappa}\right) \\
\le\ & \frac{c^3(\delta\sigma_1^2)^2}{\kappa^2} \cdot \frac{-405}{\kappa} \le 0.
\end{aligned}
$$

Next,

$$\frac{-2(1+\alpha)(2\alpha-3)c(\delta\sigma_1^2)l^2}{\kappa^3} - \frac{2c(\delta\sigma_1^2)l(3-\alpha)(1-\alpha^2)}{\kappa^2}$$

$$\leq \frac{2(1+\alpha)c(\delta\sigma_1^2)l}{\kappa^2}\left(\frac{-(2\alpha-3)l}{\kappa} - (3-\alpha)(1-\alpha)\right)$$

$$\leq \frac{2(1+\alpha)c(\delta\sigma_1^2)l}{\kappa^2}\left(\frac{3l}{\kappa} - 2(1-\alpha)\right)$$

$$\leq \frac{2(1+\alpha)c(\delta\sigma_1^2)l}{\kappa^2}\left(\frac{3l}{\kappa} - \frac{2\cdot 450}{\kappa}\right)$$

$$\leq \frac{2(1+\alpha)c(\delta\sigma_1^2)l}{\kappa^2}\left(\frac{3\cdot 27}{\kappa} - \frac{2\cdot 450}{\kappa}\right) \leq 0.$$

In both these cases, we used the fact that $\alpha \leq 1 - \frac{450}{\kappa}$ implying $-(1-\alpha) \leq \frac{-450}{\kappa}$. Finally, other remaining terms are negative. $\qquad\square$

Before rounding up the proof of the proposition, we need the following lemma to ensure that our lower bounds on the largest eigenvalue of $\mathcal{B}$ indeed affect the algorithm's rates and are true *irrespective* of where the algorithm is begun. Note that this allows our result to be *much stronger* than typical optimization lowerbounds that rely on specific initializations to ensure a component along the largest eigendirection of the update operator, for which bounds are proven.

**Lemma 13.** *For any starting iterate $\mathbf{w}_0 \neq \mathbf{w}^*$, the HB method produces a non-zero component along the largest eigen direction of $\mathcal{B}$.*

*Proof.* We note that in a similar manner as other proofs, it suffices to argue for each dimension of the problem separately. But before we start looking at each dimension separately, let us consider the $j^{\text{th}}$ dimension, and detail the approach we use to prove the claim: the idea is to examine the subspace spanned by covariance $\mathbb{E}\left[\boldsymbol{\theta}_{\cdot}^{(j)} \otimes \boldsymbol{\theta}_{\cdot}^{(j)}\right]$ of the iterates $\boldsymbol{\theta}_0^{(j)}, \boldsymbol{\theta}_1^{(j)}, \boldsymbol{\theta}_2^{(j)}, ...,$ for every starting iterate $\boldsymbol{\theta}_0^{(j)} \neq [0,0]^\top$ and prove that the largest eigenvector of the expected operator $\mathcal{B}^{(j)}$ is not orthogonal to this subspace. This implies that there exists a non-zero component of $\mathbb{E}\left[\boldsymbol{\theta}_{\cdot}^{(j)} \otimes \boldsymbol{\theta}_{\cdot}^{(j)}\right]$ in the largest eigen direction of $\mathcal{B}^{(j)}$, and this decays at a rate that is at best $\lambda_{\max}(\mathcal{B}^{(j)})$.

Since $\mathcal{B}^{(j)} \in \mathbb{R}^{4\times 4}$, we begin by examining the expected covariance spanned by the iterates $\boldsymbol{\theta}_0^{(j)}, \boldsymbol{\theta}_1^{(j)}, \boldsymbol{\theta}_2^{(j)}, \boldsymbol{\theta}_3^{(j)}$. Let $\mathbf{w}_0^{(j)} - (\mathbf{w}^*)^{(j)} = \mathbf{w}_{-1}^{(j)} - (\mathbf{w}^*)^{(j)} = k^{(j)}$. Now, this implies $\boldsymbol{\theta}_0^{(j)} = k^{(j)} \cdot [1,1]^\top$. Then,

$$\boldsymbol{\theta}_1^{(j)} = k^{(j)}\widehat{\mathbf{A}}_1^{(j)}\begin{bmatrix}1\\1\end{bmatrix}, \text{with } \widehat{\mathbf{A}}_1^{(j)} = \begin{bmatrix}1+\alpha-\delta\widehat{\mathbf{H}}_1^{(j)} & -\alpha\\ 1 & 0\end{bmatrix}, \text{ where } \widehat{\mathbf{H}}_1^{(j)} = (a_1^{(j)})^2.$$

This implies that $k$ just appears as a scale factor. This in turn implies that in order to analyze the subspace spanned by the covariance of iterates $\boldsymbol{\theta}_0^{(j)}, \boldsymbol{\theta}_1^{(j)}, ...,$ we can assume $k^{(j)} = 1$ without any loss in generality. This implies, $\boldsymbol{\theta}_0^{(j)} = [1,1]^\top$. Note that with this in place, we see that we can now drop the superscript $j$ that represents the dimension, since the analysis decouples across the dimensions $j \in \{1,2\}$. Furthermore, let the entries of the vector $\boldsymbol{\theta}_k$ be represented as $\boldsymbol{\theta}_k \overset{\text{def}}{=} [\theta_{k1} \quad \theta_{k2}]^\top$ Next, denote $1+\alpha-\delta\widehat{\mathbf{H}}_k = \hat{t}_k$. This implies,

$$\widehat{\mathbf{A}}_k = \begin{bmatrix}\hat{t}_k & -\alpha\\ 1 & 0\end{bmatrix}.$$

Furthermore,

$$\boldsymbol{\theta}_1 = \widehat{\mathbf{A}}_1\boldsymbol{\theta}_0 = \begin{bmatrix}\hat{t}_1 - \alpha\\ 1\end{bmatrix}, \quad \boldsymbol{\theta}_2 = \widehat{\mathbf{A}}_2\boldsymbol{\theta}_1 = \begin{bmatrix}\hat{t}_2(\hat{t}_1-\alpha)-\alpha\\ \hat{t}_1-\alpha\end{bmatrix},$$

$$\boldsymbol{\theta}_3 = \widehat{\mathbf{A}}_3\boldsymbol{\theta}_2 = \begin{bmatrix}\hat{t}_3(\hat{t}_2(\hat{t}_1-\alpha)-\alpha)-\alpha(\hat{t}_1-\alpha)\\ \hat{t}_2(\hat{t}_1-\alpha)-\alpha\end{bmatrix}. \tag{4}$$

Let us consider the vectorized form of $\mathbf{\Phi}_j = \mathbb{E}\left[\boldsymbol{\theta}_j \otimes \boldsymbol{\theta}_j\right]$, and we denote this as $\mathrm{vec}(\mathbf{\Phi}_j)$. Note that $\mathrm{vec}(\mathbf{\Phi}_j)$ makes $\mathbf{\Phi}_j$ become a column vector of size $4 \times 1$. Now, consider $\mathrm{vec}(\mathbf{\Phi}_j)$ for $j = 0, 1, 2, 3$ and concatenate these to form a matrix that we denote as $\mathcal{D}$, i.e.

$$\mathcal{D} = \begin{bmatrix} \mathrm{vec}(\mathbf{\Phi}_0) & \mathrm{vec}(\mathbf{\Phi}_1) & \mathrm{vec}(\mathbf{\Phi}_2) & \mathrm{vec}(\mathbf{\Phi}_3) \end{bmatrix}.$$

Now, since we note that $\mathbf{\Phi}_j$ is a symmetric $2 \times 2$ matrix, $\mathcal{D}$ should contain two identical rows implying that it has an eigenvalue that is zero and a corresponding eigenvector that is $\begin{bmatrix} 0 & -1/\sqrt{2} & 1/\sqrt{(2)} & 0 \end{bmatrix}^{\top}$. It turns out that this is also an eigenvector of $\mathcal{B}$ with an eigenvalue $\alpha$. Note that $\det(\mathcal{B}) = \alpha^4$. This implies there are two cases that we need to consider: (i) when all eigenvalues of $\mathcal{B}$ have the same magnitude $(= \alpha)$. In this case, we are already done, because there exists at least one non zero eigenvalue of $\mathcal{D}$ and this should have some component along one of the eigenvectors of $\mathcal{B}$ and we know that all eigenvectors have eigenvalues with a magnitude equal to $\lambda_{\max}(\mathcal{B})$. Thus, there exists an iterate which has a non-zero component along the largest eigendirection of $\mathcal{B}$. (ii) the second case is the situation when we have eigenvalues with different magnitudes. In this case, note that $\det(\mathcal{B}) = \alpha^4 < (\lambda_{\max}(\mathcal{B}))^4$ implying $\lambda_{\max}(\mathcal{B}) > \alpha$. In this case, we need to prove that $\mathcal{D}$ spans a three-dimensional subspace; if it does, it contains a component along the largest eigendirection of $\mathcal{B}$ which will round up the proof. Since we need to understand whether $\mathcal{D}$ spans a three dimensional subspace, we can consider a different (yet related) matrix, which we call $\mathcal{R}$ and this is defined as:

$$\mathcal{R} \overset{\text{def}}{=} \mathbb{E}\left( \begin{bmatrix} \theta_{01}^2 & \theta_{11}^2 & \theta_{21}^2 \\ \theta_{01}\theta_{02} & \theta_{11}\theta_{12} & \theta_{21}\theta_{22} \\ \theta_{02}^2 & \theta_{12}^2 & \theta_{22}^2 \end{bmatrix} \right)$$

Given the expressions for $\{\boldsymbol{\theta}_j\}_{j=0}^3$ (by definition of $\boldsymbol{\theta}_0$ and using equation 4), we can substitute to see that $\mathcal{R}$ has the following expression:

$$\mathcal{R} = \begin{bmatrix} 1 & \mathbb{E}\left[(\hat{t}_1 - \alpha)^2\right] & \mathbb{E}\left[(\hat{t}_2(\hat{t}_1 - \alpha) - \alpha)^2\right] \\ 1 & \mathbb{E}\left[\hat{t}_1 - \alpha\right] & \mathbb{E}\left[((\hat{t}_2(\hat{t}_1 - \alpha) - \alpha))(\hat{t}_1 - \alpha)\right] \\ 1 & 1 & \mathbb{E}\left[(\hat{t}_1 - \alpha)^2\right] \end{bmatrix}.$$

If we compute and prove that $\det(\mathcal{R}) \neq 0$, we are done since that implies that $\mathcal{R}$ has three non-zero eigenvalues.

This implies, we first define the following: let $q_\gamma = (t - \gamma)^2 + (c-1)x^2$. Then, $\mathcal{R}$ can be expressed as:

$$\det(\mathcal{R}) = \det\left( \begin{bmatrix} 1 & q_\alpha & q_0 q_\alpha - 2\alpha t(t - \alpha) + \alpha^2 \\ 1 & t - \alpha & t q_\alpha - \alpha(t - \alpha) \\ 1 & 1 & q_\alpha \end{bmatrix} \right)$$

$$= \det\left( \begin{bmatrix} 1 & q_\alpha & q_\alpha(q_0 - q_\alpha) - 2\alpha t(t - \alpha) + \alpha^2 \\ 1 & t - \alpha & t q_\alpha - \alpha(t - \alpha) - (t - \alpha) q_\alpha \\ 1 & 1 & 0 \end{bmatrix} \right)$$

$$= \det\left( \begin{bmatrix} 1 & q_\alpha - 1 & q_\alpha(q_0 - q_\alpha) - 2\alpha t(t - \alpha) + \alpha^2 \\ 1 & t - \alpha - 1 & t q_\alpha - \alpha(t - \alpha) - (t - \alpha) q_\alpha \\ 1 & 0 & 0 \end{bmatrix} \right)$$

$$= \det\left( \begin{bmatrix} 0 & q_\alpha - 1 & q_\alpha(q_0 - q_\alpha) - 2\alpha t(t - \alpha) + \alpha^2 \\ 0 & t - \alpha - 1 & t q_\alpha - \alpha(t - \alpha) - (t - \alpha) q_\alpha \\ 1 & 0 & 0 \end{bmatrix} \right)$$

Note: (i) $q_\alpha - 1 = (t - \alpha)^2 - 1 + (c-1)x^2 = (1 - x)^2 - 1 + (c-1)x^2 = -2x + x^2 + (c-1)x^2 = -2x + cx^2$.
(ii) $t - \alpha - 1 = -x$
(iii) $\alpha(q_\alpha - (t - \alpha)) = \alpha((t - \alpha)^2 - (t - \alpha) + (c-1)x^2) = \alpha((1 - x)(-x) + (c-1)x^2) = \alpha x(-1 + cx)$
(iv) $q_0 - q_\alpha = t^2 - (t - \alpha)^2 = \alpha(2t - \alpha) = 2t\alpha - \alpha^2$.
Then,

$$(2\alpha t - \alpha^2)q_\alpha - 2\alpha t(t - \alpha) + \alpha^2 = 2t\alpha(q_\alpha - (t - \alpha)) + \alpha^2(1 - q_\alpha)$$
$$= 2t\alpha(-x + cx^2) - \alpha^2(-2x + cx^2)$$

$$= -2t\alpha x + 2x\alpha^2 + 2t\alpha cx^2 - c\alpha^2 x^2$$
$$= 2\alpha x(-t + \alpha) + c\alpha x^2(2t - \alpha)$$
$$= -2\alpha x(1 - x) + 2c\alpha x^2(1 - x) + c\alpha^2 x^2$$
$$= 2\alpha x(1 - x)(-1 + cx) + c\alpha^2 x^2.$$

Then,

$$\det(\mathcal{R}) = \det\left(\begin{bmatrix} 0 & x(cx - 2) & 2\alpha x(1 - x)(-1 + cx) + c\alpha^2 x^2 \\ 0 & -x & \alpha x(cx - 1) \\ 1 & 0 & 0 \end{bmatrix}\right)$$

$$= x^2\alpha\det\left(\begin{bmatrix} 0 & (cx - 2) & c\alpha x + 2(1 - x)(cx - 1) \\ 0 & -1 & cx - 1 \\ 1 & 0 & 0 \end{bmatrix}\right)$$

$$= x^3\alpha\det\left(\begin{bmatrix} 0 & c & c\alpha - 2(cx - 1) \\ 0 & -1 & cx - 1 \\ 1 & 0 & 0 \end{bmatrix}\right)$$

Then,

$$\det(\mathcal{R}) = x^3\alpha\left(c(-1 + cx) - 2(-1 + cx) + c\alpha\right)$$

$$= \alpha x^3\left((c - 2)(-1 + cx) + c\alpha\right)$$

Note that this determinant can be zero when

$$\alpha = \frac{(c - 2)(1 - cx)}{c}. \tag{5}$$

We show this is not possible by splitting our argument into two parts, one about the convergent regime of the algorithm (where, $\delta\sigma_1^2 < \frac{2(1-\alpha^2)}{c+(c-2)\alpha}$) and the other about the divergent regime.

Let us first provide a proof for the convergent regime of the algorithm. For this regime, let the chosen $\delta$ be represented as $\delta^+$. Now, for the smaller eigen direction, $x = \delta^+\lambda_{\min} = c\delta^+\sigma_1^2/\kappa$. Suppose $\alpha$ was chosen as per equation 5,

$$\frac{c\alpha}{c - 2} = 1 - \frac{c^2\delta^+\sigma_1^2}{\kappa}$$
$$\implies \delta^+\sigma_1^2 = \frac{\kappa}{c^2} - \frac{\kappa\alpha}{c(c - 2)}.$$

We will now prove that $\delta^+\sigma_1^2 = \frac{\kappa}{c}\left(\frac{1}{c} - \frac{\alpha}{c-2}\right)$ is much larger than one allowed by the convergence of the HB updates, i.e., $\delta\sigma_1^2 < \frac{2(1-\alpha^2)}{c+(c-2)\alpha} \leq \frac{2(1-\alpha^2)}{c}$. In particular, if we prove that $\frac{\kappa}{c}\left(\frac{1}{c} - \frac{\alpha}{c-2}\right) > \frac{2(1-\alpha^2)}{c}$ for any admissible value of $\alpha$, we are done.

$$\frac{\kappa}{c}\left(\frac{1}{c} - \frac{\alpha}{c - 2}\right) > \frac{2(1 - \alpha^2)}{c}$$
$$\Leftrightarrow \frac{\kappa}{c} - \frac{\kappa\alpha}{c - 2} > 2 - 2\alpha^2$$
$$\Leftrightarrow \frac{\kappa}{c} - \frac{\kappa\alpha}{c - 2} > \frac{\kappa}{c} - \frac{\kappa\alpha}{c} > 2 - 2\alpha^2$$
$$\Leftrightarrow \kappa - \kappa\alpha > 2c - 2c\alpha^2$$
$$\Leftrightarrow 2c\alpha^2 - \kappa\alpha + (\kappa - 2c) > 0.$$

The two roots of this quadratic equation are $\alpha^+ = \frac{\kappa}{2c} - 1$ and $\alpha^- = 1$. Note that $\kappa \geq \widetilde{\kappa} = c$; note that there is not much any method gains over SGD if $\kappa = \mathcal{O}(c)$. And, for any $\kappa \geq 4c$, note, $\alpha^+ > \alpha^-$, indicating that the above equation holds true if $\alpha > \alpha^+ = \frac{\kappa}{2c} - 1$ or if $\alpha < \alpha^- = 1$. The latter condition is true and hence the proposition that $\delta^+\sigma_1^2 > \frac{2(1-\alpha^2)}{c+(c-2)\alpha}$ is true.

We need to prove that the determinant does not vanish in the divergent regime for rounding up the proof to the lemma.

Now, let us consider the divergent regime of the algorithm, i.e., when, $\delta\sigma_1^2 > \frac{2(1-\alpha^2)}{c+(c-2)\alpha}$. Furthermore, for the larger eigendirection, the determinant is zero when $\delta\sigma_1^2 = \frac{1-\frac{c\alpha}{c-2}}{c} = \frac{1}{c} - \frac{\alpha}{c-2}$ (obtained by substituting $x = \delta\sigma_1^2$ in equation 5). If we show that $\frac{2(1-\alpha^2)}{c+(c-2)\alpha} > \frac{1}{c} - \frac{\alpha}{c-2}$ for all admissible values of $c$, we are done. We will explore this in greater detail:

$$\frac{2(1-\alpha^2)}{c+(c-2)\alpha} > \frac{1}{c} - \frac{\alpha}{c-2}$$

$$\Leftrightarrow 2(1-\alpha^2) \geq 1 + \frac{c-2}{c}\alpha - \frac{c}{c-2}\alpha - \alpha^2$$

$$\Leftrightarrow 1 - \alpha^2 \geq \frac{-4(c-1)}{c(c-2)}\alpha$$

$$\Leftrightarrow c^2 - 2c - \alpha^2 c^2 + 2c\alpha^2 \geq -4c\alpha + 4\alpha$$

$$\Leftrightarrow c^2(1-\alpha^2) - 2c(1 - \alpha^2 - 2\alpha) - 4\alpha \geq 0.$$

considering the quadratic in the left hand size and solving it for $c$, we have:

$$c^{\pm} = \frac{2(1-\alpha^2-2\alpha) \pm \sqrt{4(1-\alpha^2-2\alpha)^2 + 16\alpha(1-\alpha^2)}}{2(1-\alpha^2)}$$

$$= \frac{(1-\alpha^2-2\alpha) \pm \sqrt{(1-\alpha^2-2\alpha)^2 + 4\alpha(1-\alpha^2)}}{(1-\alpha^2)}$$

$$= \frac{(1-\alpha^2-2\alpha) \pm \sqrt{1 + \alpha^4 + 4\alpha^2 - 2\alpha^2 - 4\alpha + 4\alpha^3 + 4\alpha(1-\alpha^2)}}{(1-\alpha^2)}$$

$$= \frac{(1-\alpha^2-2\alpha) \pm (1+\alpha^2)}{(1-\alpha^2)}$$

This holds true iff

$$c \leq c^- = \frac{-2\alpha(1+\alpha)}{1-\alpha^2} = \frac{-2\alpha}{1-\alpha},$$

or iff,

$$c \geq c^+ = \frac{2(1-\alpha)}{1-\alpha^2} = \frac{2}{1+\alpha}.$$

Which is true automatically since $c > 2$. This completes the proof of the lemma. $\square$

We are now ready to prove Lemma 5.

*Proof of Lemma 5.* Combining Lemmas 9 and 12, we see that no matter what stepsize and momentum we choose, $\mathcal{B}^{(j)}$ has an eigenvalue of magnitude at least $1 - \frac{500}{\kappa}$ for some $j \in \{1, 2\}$. This proves the lemma. $\square$

## B EQUIVALENCE OF ALGORITHM 3 AND ASGD

We begin by writing out the updates of ASGD as written out in Jain et al. (2017), which starts with two iterates $\widehat{a}_0$ and $\widehat{d}_0$, and from time $t = 0, 1, ...T - 1$ implements the following updates:

$$\widehat{b}_t = \alpha_1\widehat{a}_t + (1-\alpha_1)\widehat{d}_t \tag{6}$$

$$\widehat{a}_{t+1} = \widehat{b}_t - \delta_1\widehat{\nabla}f_{t+1}(\widehat{b}_t) \tag{7}$$

$$\widehat{c}_t = \beta_1\widehat{b}_t + (1-\beta_1)\widehat{d}_t \tag{8}$$

$$\widehat{d}_{t+1} = \widehat{c}_t - \gamma_1\widehat{\nabla}f_{t+1}(\widehat{b}_t). \tag{9}$$

Next, we specify the step sizes $\beta_1 = c_3^2/\sqrt{\kappa\widetilde{\kappa}}$, $\alpha_1 = c_3/(c_3 + \beta)$, $\gamma_1 = \beta/(c_3\lambda_{\min})$ and $\delta_1 = 1/R^2$, where $\kappa = R^2/\lambda_{\min}$. Note that the step sizes in the paper of Jain et al. (2017) with $c_1$ in their paper set to 1 yields the step sizes above. Now, substituting equation 8 in equation 9 and substituting the value of $\gamma_1$, we have:

$$\widehat{d}_{t+1} = \beta_1 \left( \widehat{b}_t - \frac{1}{c_3\lambda_{\min}} \hat{\nabla} f_{t+1}(\widehat{b}_t) \right) + (1 - \beta_1)\widehat{d}_t$$
$$= \beta_1 \left( \widehat{b}_t - \frac{\delta\kappa}{c_3} \hat{\nabla} f_{t+1}(\widehat{b}_t) \right) + (1 - \beta_1)\widehat{d}_t. \tag{10}$$

We see that $\widehat{d}_{t+1}$ is precisely the update of the running average $\bar{w}_{t+1}$ in the ASGD method employed in this paper.

We now update $\widehat{b}_t$ to become $\widehat{b}_{t+1}$ and this can be done by writing out equation 6 at $t + 1$, i.e:

$$\widehat{b}_{t+1} = \alpha_1 \widehat{a}_{t+1} + (1 - \alpha_1)\widehat{d}_{t+1}$$
$$= \alpha_1 \left( \widehat{b}_t - \delta_1 \hat{\nabla} f_{t+1}(\widehat{b}_t) \right) + (1 - \alpha_1)\widehat{d}_{t+1}. \tag{11}$$

By substituting the value of $\alpha_1$ we note that this is indeed the update of the iterate as a convex combination of the current running average and a short gradient step as written in this paper. In this paper, we set $c_3$ to be equal to $0.7$, and any constant less than 1 works. In terms of variables, we note that $\alpha$ in this paper's algorithm description maps to $1 - \beta_1$.

## C  MORE DETAILS ON EXPERIMENTS

In this section, we will present more details on our experimental setup.

### C.1  LINEAR REGRESSION

In this section, we will present some more results on our experiments on the linear regression problem. Just as in Appendix A, it is indeed possible to compute the expected error of all the algorithms among SGD, HB, NAG and ASGD, by tracking certain covariance matrices which evolve as linear systems. For SGD, for instance, denoting $\boldsymbol{\Phi}_t^{SGD} \stackrel{\text{def}}{=} \mathbb{E}\left[\left(\mathbf{w}_t^{SGD} - w^*\right) \otimes \left(\mathbf{w}_t^{SGD} - w^*\right)\right]$, we see that $\boldsymbol{\Phi}_{t+1}^{SGD} = \mathcal{B} \circ \boldsymbol{\Phi}_t^{SGD}$, where $\mathcal{B}$ is a linear operator acting on $d \times d$ matrices such that $\mathcal{B} \circ M \stackrel{\text{def}}{=} M - \delta HM - \delta MH + \delta^2 \mathbb{E}\left[\langle x, Mx \rangle xx^\top\right]$. Similarly, HB, NAG and ASGD also have corresponding operators (see Appendix A for more details on the operator corresponding to HB). The largest magnitude of the eigenvalues of these matrices indicate the rate of decay achieved by the particular algorithm – smaller it is compared to 1, faster the decay.

We now detail the range of parameters explored for these results: the condition number $\kappa$ was varied from $\{2^4, 2^5, .., 2^{28}\}$ for all the optimization methods and for both the discrete and gaussian problem. For each of these experiments, we draw 1000 samples and compute the empirical estimate of the fourth moment tensor. For NAG and HB, we did a very fine grid search by sampling 50 values in the interval $(0, 1]$ for both the learning rate and the momentum parameter and chose the parameter setting that yielded the smallest $\lambda_{\max}(\mathcal{B})$ that is less than 1 (so that it falls in the range of convergence of the algorithm). As for SGD and ASGD, we employed a learning rate of $1/3$ for the Gaussian case and a step size of $0.9$ for the discrete case. The statistical advantage parameter of ASGD was chosen to be $\sqrt{3\kappa/2}$ for the Gaussian case and $\sqrt{2\kappa/3}$ for the Discrete case, and the a long step parameters of $3\kappa$ and $2\kappa$ were chosen for the Gaussian and Discrete case respectively. The reason it appears as if we choose a parameter above the theoretically maximal allowed value of the advantage parameter is because the definition of $\kappa$ is different in this case. The $\kappa$ we speak about for this experiment is $\lambda_{\max}/\lambda_{\min}$ unlike the condition number for the stochastic optimization problem. In a manner similar to actually running the algorithms (the results of whose are presented in the main paper), we also note that we can compute the rate as in equation 1 and join all these rates using a curve and estimate its slope (in the $\log$ scale). This result is indicated in table 3.

Figure 7 presents these results, where for each method, we did grid search over all parameters and chose parameters that give smallest $\lambda_{\max}$. We see the same pattern as in Figure 1 from actual

runs – SGD,HB and NAG all have linear dependence on condition number $\kappa$, while ASGD has a dependence of $\sqrt{\kappa}$.

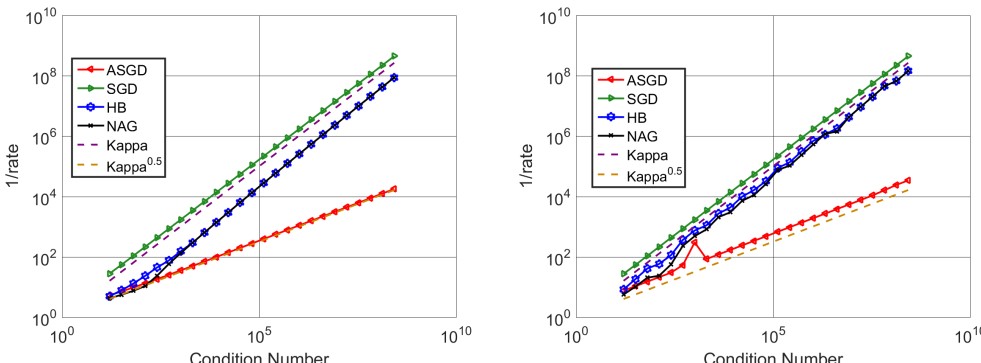

Figure 7: Expected rate of error decay (equation 1) vs condition number for various methods for the linear regression problem. Left is for discrete distribution and right is for Gaussian distribution.

| Algorithm | Slope – discrete | Slope – Gaussian |
|:---:|:---:|:---:|
| SGD | 0.9990 | 0.9995 |
| HB | 1.0340 | 0.9989 |
| NAG | 1.0627 | 1.0416 |
| ASGD | 0.4923 | 0.4906 |

Table 3: Slopes (i.e. $\gamma$) obtained by fitting a line to the curves in Figure 7. A value of $\gamma$ indicates that the error decays at a rate of $\exp\left(\frac{-t}{\kappa^\gamma}\right)$. A smaller value of $\gamma$ indicates a faster rate of error decay.

## C.2 Autoencoders for MNIST

We begin by noting that the learning rates tend to vary as we vary batch sizes, which is something that is known in theory (Jain et al., 2016). Furthermore, we extend the grid especially whenever our best parameters of a baseline method tends to land at the edge of a grid. The parameter ranges explored by our grid search are:

**Batch Size** 1: (parameters chosen by running for 20 epochs)

- SGD: learning rate: $\{0.01, 0.01\sqrt{10}, 0.1, 0.1\sqrt{10}, 1, \sqrt{10}, 5, 10, 20, 10\sqrt{10}, 40, 60, 80, 100.$
- NAG/HB: learning rate: $\{0.01\sqrt{10}, 0.1, 0.1\sqrt{10}, 1, \sqrt{10}, 10\}$, momentum $\{0, 0.5, 0.75, 0.9, 0.95, 0.97\}$.
- ASGD: learning rate: $\{2.5, 5\}$, long step $\{100.0, 1000.0\}$, advantage parameter $\{2.5, 5.0, 10.0, 20.0\}$.

**Batch Size** 8: (parameters chosen by running for 50 epochs)

- SGD: learning rate: $\{0.001, 0.001\sqrt{10.0}, 0.01, 0.01\sqrt{10}, 0.1, 0.1\sqrt{10}, 1, \sqrt{10}, 5, 10$ , $10\sqrt{10}, 40, 60, 80, 100, 120, 140\}$.
- NAG/HB: learning rate: $\{5.0, 10.0, 20.0, 10\sqrt{10}, 40, 60\}$, momentum $\{0, 0.25, 0.5, 0.75, 0.9, 0.95\}$.
- ASGD: learning rate $\{40, 60\}$. For a long step of 100, advantage parameters of $\{1.5, 2, 2.5, 5, 10, 20\}$. For a long step of 1000, we swept over advantage parameters of $\{2.5, 5, 10\}$.

C.3 DEEP RESIDUAL NETWORKS FOR CIFAR-10

In this section, we will provide more details on our experiments on cifar-10, as well as present some additional results. We used a weight decay of $0.0005$ in all our experiments. The grid search parameters we used for various algorithms are as follows. Note that the ranges in which parameters such as learning rate need to be searched differ based on batch size (Jain et al., 2016). Furthermore, we tend to extrapolate the grid search whenever a parameter (except for the learning rate decay factor) at the edge of the grid has been chosen; this is done so that we always tend to lie in the interior of the grid that we have searched on. Note that for the purposes of the grid search, we choose a hold out set from the training data and add it in to the training data after the parameters are chosen, for the final run.

**Batch Size** 8: Note: (i) parameters chosen by running for $40$ epochs and picking the grid search parameter that yields the smallest validation $0/1$ error. (ii) The validation set decay scheme that we use is that if the validation error does not decay by at least $1\%$ every three passes over the data, we cut the learning rate by a constant factor (which is grid searched as described below). The minimal learning rate to use is fixed to be $6.25 \times 10^{-5}$, so that we do not decay far too many times and curtail progress prematurely.

- SGD: learning rate: $\{0.0033, 0.01, 0.033, 0.1, 0.33\}$, learning rate decay factor $\{5, 10\}$.
- NAG/HB: learning rate: $\{0.001, 0.0033, 0.01, 0.033\}$, momentum $\{0.8, 0.9, 0.95, 0.97\}$, learning rate decay factor $\{5, 10\}$.
- ASGD: learning rate $\{0.01, 0.0330, 0.1\}$, long step $\{1000, 10000, 50000\}$, advantage parameter $\{5, 10\}$, learning rate decay factor $\{5, 10\}$.

**Batch Size** 128: Note: (i) parameters chosen by running for $120$ epochs and picking the grid search parameter that yields the smallest validation $0/1$ error. (ii) The validation set decay scheme that we use is that if the validation error does not decay by at least $0.2\%$ every four passes over the data, we cut the learning rate by a constant factor (which is grid searched as described below). The minimal learning rate to use is fixed to be $1 \times 10^{-3}$, so that we do not decay far too many times and curtail progress prematurely.

- SGD: learning rate: $\{0.01, 0.03, 0.09, 0.27, 0.81\}$, learning rate decay factor $\{2, \sqrt{10}, 5\}$.
- NAG/HB: learning rate: $\{0.01, 0.03, 0.09, 0.27\}$, momentum $\{0.5, 0.8, 0.9, 0.95, 0.97\}$, learning rate decay factor $\{2, \sqrt{10}, 5\}$.
- ASGD: learning rate $\{0.01, 0.03, 0.09, 0.27\}$, long step $\{100, 1000, 10000\}$, advantage parameter $\{5, 10, 20\}$, learning rate decay factor $\{2, \sqrt{10}, 5\}$.

As a final remark, for any comparison across algorithms, such as, (i) ASGD vs. NAG, (ii) ASGD vs HB, we fix the starting learning rate, learning rate decay factor and decay schedule chosen by the best grid search run of NAG/HB respectively and perform a grid search over the long step and advantage parameter of ASGD. In a similar manner, when we compare (iii) SGD vs NAG or, (iv) SGD vs. HB, we choose the learning rate, learning rate decay factor and decay schedule of SGD and simply sweep over the momentum parameter of NAG or HB and choose the momentum that offers the best validation error.

We now present plots of training function value for different algorithms and batch sizes.

**Effect of minibatch sizes**: Figure 8 plots training function value for batch sizes of $128$ and $8$ for SGD, HB and NAG. We notice that in the initial stages of training, NAG obtains substantial improvements compared to SGD and HB for batch size $128$ but not for batch size $8$. Towards the end of training however, NAG starts decreasing the training function value rapidly for both the batch sizes. The reason for this phenomenon is not clear. Note however, that at this point, the test error has already stabilized and the algorithms are just overfitting to the data.

**Comparison of ASGD with momentum methods**: We now present the training error plots for ASGD compared to HB and NAG in Figures 9 and 10 respectively. As mentioned earlier, in order to see a clear trend, we constrain the learning rate and decay schedule of ASGD to be the same as that of HB and NAG respectively, which themselves were learned using grid search. We see

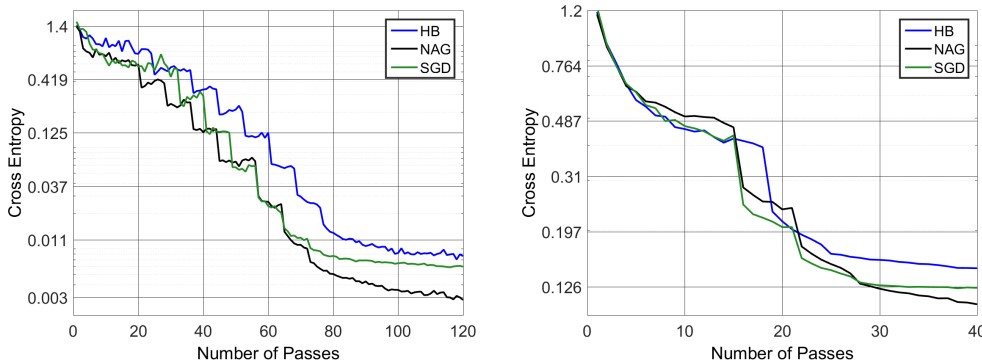

Figure 8: Training loss for batch sizes 128 and 8 respectively for SGD, HB and NAG.

similar trends as in the validation error plots from Figures 5 and 6. Please see the figures and their captions for more details.

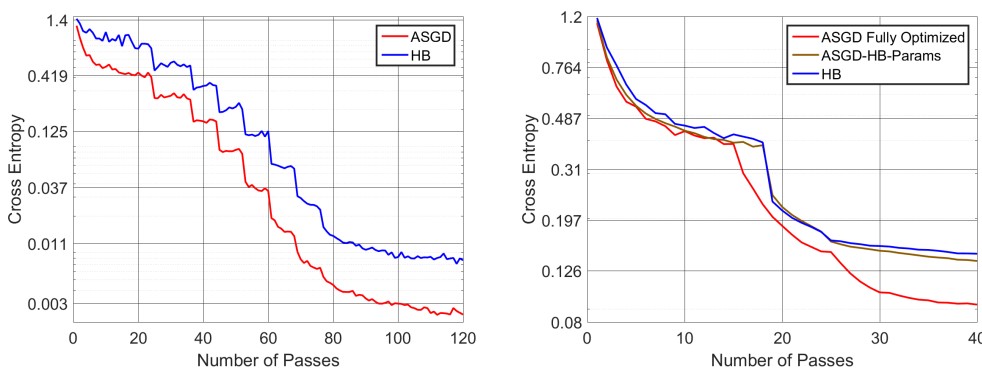

Figure 9: Training function value for ASGD compared to HB for batch sizes 128 and 8 respectively.

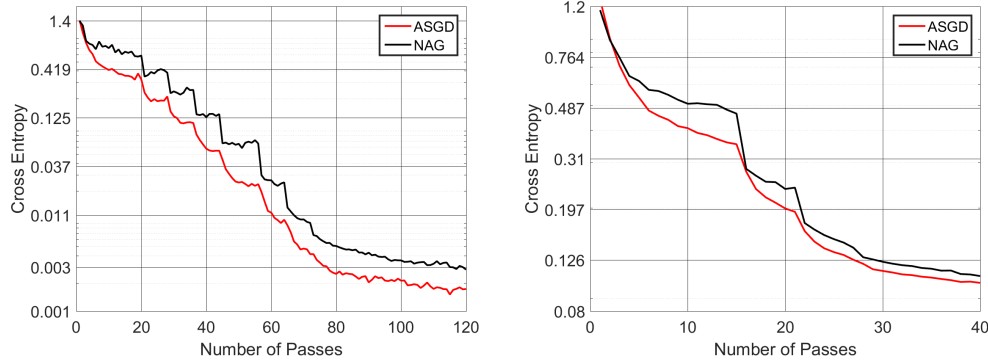

Figure 10: Training function value for ASGD compared to NAG for batch size 128 and 8 respectively.

