# OpenReview forum: "On the insufficiency of existing momentum schemes for Stochastic Optimization"
_ICLR.cc/2018/Conference — Accept (Oral)_

### Official Review · AnonReviewer2 · 2017-11-25
**Nice idea, Like the paper**

**Rating:** 7
**Confidence:** 4

**Review:**

I like the idea of the paper. Momentum and accelerations are proved to be very useful both in deterministic and stochastic optimization. It is natural that it is understood better in the deterministic case. However, this comes quite naturally, as deterministic case is a bit easier ;) Indeed, just recently people start looking an accelerating in stochastic formulations. There is already accelerated SVRG, Jain et al 2017, or even Richtarik et al (arXiv: 1706.01108, arXiv:1710.10737).

I would somehow split the contributions into two parts:
1) Theoretical contribution: Proposition 3 (+ proofs in appendix)
2) Experimental comparison.

I like the experimental part (it is written clearly, and all experiments are described in a lot of detail).

I really like the Proposition 3 as this is the most important contribution of the paper. (Indeed, Algorithms 1 and 2 are for reference and Algorithm 3 was basically described in Jain, right?).

Significance: I think that this paper is important because it shows that the classical HB method cannot achieve acceleration in a stochastic regime.

Clarity: I was easy to read the paper and understand it.

Few minor comments:
1. Page 1, Paragraph 1: It is not known only for smooth problems, it is also true for simple non-smooth (see e.g. https://link.springer.com/article/10.1007/s10107-012-0629-5)
2. In abstract : Line 6 - not completely true, there is accelerated SVRG method, i.e. the gradient is not exact there, also see Recht (https://arxiv.org/pdf/1701.03863.pdf) or Richtarik et al (arXiv: 1706.01108, arXiv:1710.10737) for some examples where acceleration can be proved when you do not have an exact gradient.
3. Page 2, block "4" missing "." in "SGD We validate"....
4. Section 2. I think you are missing 1/2 in the definition of the function. Otherwise, you would have a constant "2" in the Hessian, i.e. H= 2 E[xx^T]. So please define the function as  f_i(w) = 1/2 (y - <w,x_i>)^2. The same applies to Section 3.
5. Page 6, last line, .... was downloaded from "pre". I know it is a link, but when printed, it looks weird.

---

> ### Author Response · Authors · 2017-12-20
> **Rebuttal**
>
> Thanks a lot for insightful comments.  We have updated the paper taking into account several of your comments. We will make more updates according to your suggestions.
>
>
> Paper organization: we will try to better organize the paper to highlight the contributions.
> Proposition 3's importance: yes, your assessment is spot on.
>
> Minor comment 1,2: Thanks for pointing the minor mistake, we have updated the corresponding lines. Papers such as Accelerated SVRG, Recht et al. are offline stochastic accelerated methods. The paper of Richtarik (arXiv:1706.01108) deals with solving consistent linear systems in the offline setting; (arXiv:1710.10737) is certainly relevant and we will add more detailed comparison with this line of work.
> Minor comment 3, 5: thanks for pointing out  the typos. They are fixed.
> Minor comment 4: Actually, the problem is a discrete problem where one observes one hot vectors in 2-dimensions, each of the vectors can occur with probability 1/2. So this is the reason why the Hessian does not carry an added factor of 2.

---

### Official Review · AnonReviewer1 · 2017-11-28
**Good paper, accept**

**Rating:** 7
**Confidence:** 3

**Review:**

I wonder how the ASGD compares to other optimization schemes applicable to DL, like Entropy-SGD, which is yet another algorithm that provably improves over SGD. This question is also valid when it comes to other optimization schemes that are designed for deep learning problems. For instance, Entropy-SGD and Path-SGD should be mentioned and compared with. As a consequence, the literature analysis is insufficient.

Authors provided necessary clarifications. I am raising my score.

---

> ### Author Response · Authors · 2017-12-20
> **Rebuttal**
>
> Thanks for your comments.
>
> We have cited Entropy SGD and Path SGD papers and discuss the differences in Section 6 (related works). However, both the methods are complementary to our method.
>
> Entropy SGD adds a local strong convexity term to the objective function to improve generalization. However, currently we do not understand convergence rates or generalization performance of the technique rigorously, even for convex problems. The paper proposes to use SGD to optimize the altered objective function and mentions that one can use SGD+momentum as well (below algorithm box on page 6). Naturally, one can use the ASGD method as well to optimize the proposed objective function in the paper.
>
> Path SGD uses a modified SGD like update to ensure invariance to the scale of the data. Here again, the main goal is orthogonal to our work and one can easily use ASGD method in the same framework.

---

### Official Review · AnonReviewer3 · 2017-12-13
**Accept**

**Rating:** 8
**Confidence:** 5

**Review:**

I only got access to the paper after the review deadline; and did not have a chance to read it until now. Hence the lateness and brevity.

The paper is reasonably well written, and tackles an important problem. I did not check the mathematics.

Besides the missing literature mentioned by other reviewers (all directly relevant to the current paper), the authors should also comment on the availability of accelerated methods inn the finite sum / ERM setting. There, the questions this paper is asking are resolved, and properly modified stochastic methods exist which offer acceleration over SGD (and not through minibatching). This paper does not comment on these developments. Look at accelerated SDCA (APPROX, ASDCA), accelerated SVRG (Katyusha) and so on.

Provided these changes are made, I am happy to suggest acceptance.

---

> ### Author Response · Authors · 2017-12-20
> **Rebuttal**
>
> Thanks for the references, we have included them in the paper and added a paragraph in Section 6 providing detailed comparison and key differences that we summarize below:
>
> ASDCA, Katyusha, accelerated SVRG: these methods are "offline" stochastic algorithms that is they require  multiple passes over the data and require multiple rounds of full gradient computation (over the entire training data). In contrast, ASGD is a single pass algorithm and requires gradient computation only a single data point at a time step. In the context of deep learning, this is a critical difference, as computing gradient over entire training data can be extremely slow. See Frostig, Ge, Kakade, Sidford ``Competing with the ERM in a single pass" (https://arxiv.org/pdf/1412.6606.pdf) for a more detailed discussion on online vs offline stochastic methods.
>
> Moreover, the rate of convergence of the ASDCA depend on \sqrt{\kappa n} while the method studied in this paper has \sqrt{\kappa \tilde{kappa}} dependence where \tilde{kappa} can be much smaller than n.

---

### Author Response · Authors · 2018-01-05
**list of changes made to the manuscript**

We group the list of changes made to the manuscript based on suggestions of reviewers:

AnonReviewer 3:
- Added a paragraph on accelerated and fast methods for finite sums and their implications in the deep learning context. (in related work)

AnonReviewer 2:
- Included reference on Acceleration for simple non-smooth problems. (in page 1)
- Included reference on Accelerated SVRG and other suggested references. (in related work)
- Fixed citations for pytorch/download links and fixed typos.

AnonReviewer 1:
- Added a paragraph on entropic sgd and path normalized sgd and their complimentary nature compared to this work's message (in related work section).

Other changes:
- In the related work: background about Stochastic Heavy Ball, adding references addressing reviewer feedback.
- Removed statement on generalization/batch size. (page 2)
- Fixed minor typos. (page 3)
- Added comment about NAG lower bound conjecture. (page 4, below proposition 3)

---

### Public Comment · ~James_Martens1 · 2018-02-22
**The parameters of momentum methods**

Looking at the update equations for ASGD they seem like they would produce updates that are some kind of exponentially decayed average of gradients, just like HB or NAG.  However the rate of this decay, as well as the effective learning, might vary across iterations.

Here it becomes crucial what you consider to the parameters of each optimizer.   If you think of them as defining the traditional momentum decay and learning rate parameters (as these appear in standard packages like TensorFlow), then the average of gradients will decay at a fixed rate.  But if you think of them as defining a schedule over the traditional momentum decay and learning rate parameters, as was done in Sutskever et al. (2013) (in a way inspired by Nesterov's original prescriptions for NAG), the effective decay rate will change over time.

Did you experiments take the parameters to be the learning rate and momentum decay constant for both SGD and NAG?

The picture becomes even more complicated if you add Polyak averaging into this mix.  The problem you study in your theory is unusual because SGD with a fixed learning rate actually converges on it (as 1/exp(t)) without Polyak average (or learning rate decay).  (This is because an error of 0 is obtainable on each individual training case, which means that the stochastic gradient can itself converge to 0 as w approaches w*.)  Nonetheless I wonder what the effect of averaging would be on this problem if it were layered on top of SGD or NAG, and if it would help bridge the gap between those methods and ASGD in this particular case.

---

> ### Author Response · Authors · 2018-02-28
> **re: The parameters of momentum methods**
>
> Thank you for your interest and questions:
>
> [1] Regarding decaying momentum: In smooth convex optimization, Nesterov's scheme is employed with time-varying momentum. In the smooth+strongly convex case, any accelerated method (including Nesterov's method) is implemented with a constant momentum term (see for example, Bubeck 2015). So, in a sense, for the (strongly convex) consistent linear system case described in the paper, a constant momentum term is in accordance with the prescription from convex optimization.
>
> For the neural net examples, we used typical strategies of a constant momentum term, as employed in common packages (like tensorflow/pytorch), since these appear to be the most widely used in practice. Moreover, there are several parameters to tune and perform a grid search on, so a scheme for varying momentum just adds to making these grid searches longer. We note that time-varying momentum schemes can also be added to the proposed ASGD method and these comparisons can be made.
>
> Finally, in the case that you'd think that the ASGD method as described in the paper varies the rate of decay of average gradients over iterations, we would like to clarify that this is not the case and that ASGD also retains constant learning rate/momentum parameters across iterations.
>
> [2] For the bounds with Polyak averaging: Note that averaging the iterates of any stochastic gradient method provides gains only when there is additive noise. In the noiseless case, as you mention, the iterates of SGD converge linearly to the minimizer. Averaging the iterates is strictly worse than the final iterate for the noiseless case and leads to sub-linear convergence of the iterates towards the minimizer. So by no means, averaging iterates of SGD/NAG/HB can improve over ASGD (or even unaveraged versions of SGD/NAG/HB for that matter). For the behavior of Polyak-averaged SGD, refer to Jain et al. 2016 ''Parallelizing stochastic approximation through mini-batching and tail-averaging'' - Figure 2a (red and green curves represent averaged and unaveraged SGD respectively) and Theorem 2 for theoretical bounds (by setting $\Sigma=0$ for the consistent linear system case).

---

> > ### Public Comment · ~James_Martens1 · 2018-02-28
> > **re: The parameters of momentum methods**
> >
> > [1]
> > I think you may have misunderstood my first point.
> >
> > I was saying that looking at your update equations it seems plausible that they correspond to standard momentum or Nesterov's method, but with a learning rate and momentum constant (as they are defined on those methods) that change over time.  It's actually hard to write down a first order method where this isn't true.  To be clear, I'm NOT saying that your method changes its own parameters over time.  I'm saying that your method, with fixed parameters, might correspond to the more classic methods with time varying parameters.
> >
> > As far as performance on neural nets goes, I'm somewhat of an expert on this topic and in my experience time-varying momentum is crucial for state-of-the-art optimization performance on these kinds of autoencoder tasks.  A high value for the momentum can cause a lot more instability early in optimization vs later one, which is why one should always start low and increase over time.  What is implemented by default in Tensorflow or PyTorch, or what certain convex optimization theory papers use to prove certain asymptotic upper bounds, is a different matter.
> >
> > [2]
> > I'm not sure what you mean by "the noiseless case".  Do you mean deterministic optimization?  Then SGD is just GD.  Is it not the case that your are sampling stochastic gradients for this objective function using these 'a' and 'b' random variables?
> >
> > I was pointing out that your problem is unusual because the stochastic gradients converge to zero (not just their expectation) as the parameters converge to the optimum.   This is the "realizable case", where the model can perfectly capture the data-generating process.  If your problem doesn't have this property then SGD with Polyak averaging is asymptotically optimal.   Do you know how well your method would perform in the non-realizable case?   Your paper would be clearer if you were explicit about this.  You could easily make the argument that "we are better than Polyak averaging in the realizable case", and this would be interesting by itself, even if the performance in the non-realizable case were worse.
> >
> > EDIT:  I misused "realizable" above.  You're right that it's actually a weaker condition than having the gradients all converge to zero.  In general it means that the model (in this case a linear one) can capture the true data distribution.  This agrees with the definition you gave below for the linear model case.

---

> > > ### Author Response · Authors · 2018-03-07
> > > **re: The parameters of momentum methods**
> > >
> > > Part 1 of 2:
> > > [1] Thanks for clarifying your precise question. Answer: ASGD is indeed a generalization of NAG (i.e., there is some setting of parameters of ASGD which recovers NAG with fixed step size and momentum parameters) but ASGD does not correspond to NAG with time-varying step size/momentum. Below, we clarify the precise difference between NAG and ASGD (both with fixed parameters) and intuitively why ASGD might perform better than NAG.
> > >
> > > ASGD
> > > ----
> > > Start at x_0=y_0=v_0 (say).
> > > For t = 1,2,...,T Repeat:
> > >     v_t = (1-alpha) * (y_{t-1} - gamma * g(y_{t-1})) + alpha * v_{t-1};   /* gradient descent with long step "gamma" and exponential decaying average of past gradients*/
> > >     y_t = beta * (y_{t-1} - delta * g(y_{t-1})) + (1-beta) * v_t;	/* gradient descent with short step "delta" and exponential decaying average of past gradients*/
> > > end
> > >
> > > where g(y_{t-1}) is the gradient at y_{t-1} and "alpha", "gamma" and "delta" are the parameters of ASGD and "beta = c/(c+1-alpha)" (0<c<1 is arbitrary; we choose c = 0.7). Of these parameters, "delta" corresponds to the step size in standard terminology. This algorithm performs an exponentially decaying average of past gradients with long step "gamma" and short step "delta". The long step "gamma" is of size "1/mu" and short step "delta" is of size "1/L", where "mu" and "L" are strong convexity and smoothness of the problem respectively.
> > >
> > > If we set "alpha = theta*(1+c)/(c+theta)" and "gamma = (delta/(1-alpha)^2) * c * (1+c*theta)/(c+theta)", then ASGD written above is exactly NAG with step size "delta" and momentum "theta". To see this, we note that the variables "y_t" and "x_t = y_{t-1} - delta * g(y_{t-1})" from ASGD above satisfy
> > >
> > >     x_t = y_{t-1} - delta * g(y_{t-1});
> > >     y_t = (1+theta) * x_t - theta * x_{t-1};
> > >
> > > which correspond to NAG updates. In this setting, the decay factor "(1-alpha) ~ \sqrt(delta/gamma)" since for 0<theta<1, c * (1+c*theta)/(c+theta) is some reasonable constant between 0 and 2. For "delta = 1/L" and "gamma = 1/mu", we have "(1-alpha) ~ \sqrt(mu/L)".
> > >
> > > The difference in ASGD however, is that the decay factor "1-alpha" can be much smaller than "\sqrt(delta/gamma)" since it is an independent parameter. In fact, there are some bad problems, where "1-alpha" needs to be smaller than "delta/gamma" (otherwise the algorithm might diverge) and for this choice we do not see acceleration (in these cases, note that "1-alpha ~ mu/L"). On the other hand, for good problems, "1-alpha" can be chosen to be closer to "\sqrt(delta/gamma)" and for this choice we do see acceleration. See Jain et al. 2017 (https://arxiv.org/abs/1704.08227) for examples of such good problems (where acceleration is possible in the stochastic world) and bad problems (where acceleration is not possible in the stochastic world).
> > >
> > > This view also suggests a plausible explanation for why time varying momentum parameter is perhaps necessary to get good performance for NAG (on several problems with stochastic gradients). For NAG to be stable and not diverge on some problems, we might require "1-alpha ~ delta/gamma" while NAG by design enforces "1-alpha ~\sqrt(delta/gamma)". This means that "delta/gamma ~ \sqrt(delta/gamma)" or equivalently "\sqrt(delta/gamma) ~ 1". This implies that "theta" is away from 1 (cannot use large momentum). ASGD overcomes this issue by decoupling the short step-long step ratio ("delta/gamma") from the decay factor and using appropriate decay factors to ensure convergence of the algorithm.
> > >
> > > To summarize the discussion, ASGD is indeed a generalization of NAG by decoupling the decay factor for average gradients from short step-long step ratio. However, the decay factor/short step/long step do not change with time. In our view, this seems to fix NAG by making it convergent with larger "long steps" as compared to vanilla NAG.
> > >
> > > It would be very interesting to further try varying the parameters of ASGD for the neural net experiments and verify if it indeed improves the performance of ASGD. For the theoretical example mentioned in the paper, this is not required.

---

> > > ### Author Response · Authors · 2018-03-07
> > > **re: The parameters of momentum methods**
> > >
> > > Part 2 of 2:
> > > [2] There appears to be some discrepancy in the usage of realizable and agnostic cases. We will clarify these issues more precisely:
> > >
> > > Let b = w.a + eps be underlying data generation model, with w.a = sum_i w_i a_i;
> > >
> > > "Noiseless case" -- In this case, eps = 0 always, we have the consistent linear system case since there exists w* such that for all (a,b), b = a.w*; This is the case considered in this paper. However, this does not mean that we have standard gradient descent since we only get gradient information from a single sample. This setting carries what is known as "multiplicative" noise owing to sampling gradients (i.e., from sampling 'a' and 'b') instead of computing a full gradient.
> > >
> > > "Realizable case" -- In this case, eps is a zero mean random variable and independent of a. Example: sample epsilon from a zero mean gaussian with standard deviation sigma.
> > >
> > > "Non-realizable/agnostic case" -- In this case, eps shares correlations with a.
> > >
> > > Let us now give some background on the error of SGD type algorithms. The error of any SGD type algorithm can be written as a sum of two parts: "bias" representing dependence of the error on the starting point w_0, and "variance" due to the noise "eps". If we run SGD or similar methods with a fixed step size and consider the last point, the "bias" error decays geometrically as exp(-n) but the "variance" error does not decay with n (here 'n' is the number of samples or SGD steps). Polyak averaging fixes this issue and decays the "variance" error at the right 1/n rate. However, if we average the iterates right from the start, the rate of decay of "bias" becomes 1/n^2, which is sub-optimal compared to exp(-n) from before -- this is the motivation to tail-average (i.e., average only the last several iterates). This gets the best of both worlds in the sense that we get a geometric exp(-n) decay on the "bias" term and the optimal 1/n decay of the "variance" term. However, it is important to note that there is a problem dependent constant that determines the exp(-n) rate of "bias" decay. More concretely, for (tail-averaged or non-averaged) SGD the rate of "bias" decay is exp(-n/\kappa) where \kappa is the condition number. In this paper, we show that the same rate exp(-n/\kappa) is tight for both (non-averaged) HB and NAG (theoretically for HB and empirically for both). The rate of "bias" decay of Polyak-averaged or tail-averaged HB/NAG can only be worse (averaging never helps the "bias" term). Jain et al. 2017 shows that tail-averaged ASGD gets "bias" decay rate exp(-n/\sqrt{\kappa \tilde{\kappa}}), which is always better than that of SGD/HB/NAG since \tilde{\kappa} \leq \kappa. Furthermore, they also show that tail-averaged ASGD decays the "variance" error at the right 1/n rate. This means that tail-averaged ASGD improves upon the "bias" decay rate as compared to SGD/HB/NAG while achieving the same (optimal upto absolute numerical constants i.e., not problem dependent) decay rate on the "variance" term as (Polyak-averaged) SGD.
> > >
> > > So to summarize, ASGD improves upon the "bias" decay rate of SGD/HB/NAG. Polyak averaging or tail-averaging is a complementary technique and improves the "variance" decay rate. For instance, tail-averaging can be used on top of ASGD and this is better than Polyak-averaged or tail-averaged SGD.
> > >
> > > Since the improvement of ASGD over SGD/HB/NAG is in the "bias" term, we tried to illustrate this using an example where the "variance" term is equal to zero. This is the reason we consider the noiseless or consistent linear system case. While we illustrate our results in this scenario, the claims of superiority of ASGD over SGD/HB/NAG carry over to the realizable case as well (i.e., eps is a zero mean, independent random variable) due to the reasoning in the above two paragraphs.
> > >
> > > References: For a precise understanding of the behavior of SGD for least squares with realizable/agnostic noise with or without Polyak averaging, refer to "Parallelizing Stochastic Approximation Through Mini-Batching and Tail-Averaging" (https://arxiv.org/abs/1610.03774). Behavior of ASGD for least squares and Polyak averaging of the final few iterates can be precisely understood from "Accelerating Stochastic Gradient Descent" (https://arxiv.org/abs/1704.08227).

---

### Decision · Program_Chairs · 2018-01-29
**ICLR 2018 Conference Acceptance Decision**

**Decision:**

Accept (Oral)

**Comment:**

The reviewers unanimously recommended that this paper be accepted, as it contains an important theoretical result that there are problems for which heavy-ball momentum cannot outperform SGD. The theory is backed up by solid experimental results, and the writing is clear. While the reviewers were originally concerned that the paper was missing a discussion of some related algorithms (ASVRG and ASDCA) that were handled in discussion.